# Quantifying uncertainty in flood predictions due to river bathymetry estimation

Martin Nguyen[1, 2, 3], Matthew D. Wilson[1, 3], Emily M. Lane[4], James Brasington[2, 3], and Rose A. Pearson[4]

[1]Geospatial Research Institute, University of Canterbury, Christchurch, New Zealand
[2]Waterways Centre, University of Canterbury, Christchurch, New Zealand
[3]School of Earth and Environment, University of Canterbury
[4]National Institute of Water and Atmospheric Research (NIWA), Christchurch, New Zealand
**Correspondence:** Martin Nguyen (tmn52@uclive.ac.nz)

**Abstract.** River bathymetry is important for accurate flood inundation modelling but is often unavailable due to the time-intensive and expensive nature of its acquisition. This leads to several proposed and implemented approaches for its estimation. However, the errors in estimations inherent in these methods and how they affect the accuracy of the flood inundation modelling outputs, has not been extensively researched. Hence, to contribute, we investigate the sensitivity of flood predictions to the errors in river slope, width, and bank-full flow used in two formulas - the Uniform Flow and the Conceptual Multivariate Regression - for estimating river bathymetry. In this study, we employed a Monte Carlo framework to introduce random errors into these parameters drawn from a normal distribution with zero mean and a standard deviation set to 10% of their best estimates. Using this process, we generated 50 simulated river bathymetries for each parameter along with an additional 50 where the errors were applied to all parameters simultaneously. The riverbeds generated from these bathymetries were combined with topographic LiDAR data to create model grids. Each grid was used in the hydrodynamic model LISFLOOD-FP to simulate the 2005 flood event in the Waikanae River area of New Zealand. We assessed the resulting flood inundation predictions for their variability and sensitivity. The results indicate that between two methods, the errors in the parameters in the Uniform Flow formula are associated with greater uncertainty in flood inundation depths and extents compared to the Conceptual Multivariate Regression. Among the parameters, the width errors correspond to the highest uncertainty, while the slope errors correspond to the lowest.

## 1 Introduction

River bathymetry refers to the river depth measurement (Panigrahi, 2014). It plays a crucial role in flood modelling because it determines when and where water leaves the river channel and starts to flood overland (Cook and Merwade, 2009; Awadallah et al., 2022). Currently, hydrographic surveys and remote sensing methods, especially swath beam sonar and blue-green LiDAR, are prevalently employed to obtain these river bathymetric data (Costa et al., 2009; Kinzel et al., 2013; Dey et al., 2019). Multi-beam sonar is effective but time-consuming, while blue-green LiDAR is faster but does not work in sediment-laden or deep water, and both of them are expensive (Bailly et al., 2010; Flener et al., 2012; Bures et al., 2019). For

these reasons, various approaches have been proposed to estimate these data (Ghorbanidehno et al., 2021; Araújo and Hedley, 2023).

Dey et al. (2019) categorised these methods into two groups. The first one assumes rivers with simple geometric shapes like triangular (Gichamo et al., 2012; Saleh et al., 2013; Bhuyian et al., 2015), rectangular (Trigg et al., 2009; Saleh et al., 2013; Grimaldi et al., 2018), trapezoidal (Saleh et al., 2013), or parabolic (Bhuyian et al., 2015) cross-sections. Despite the fast and simple process, these assumptions might be significantly different from realistic rivers. The other group applies more complex hydraulic (Price, 2009; Bhuyian et al., 2015) and geomorphological (Brown et al., 2014) principles to create more realistic
underwater terrain. However, they require more data and heavy computation.

Some studies build up formulas to estimate river bathymetry based on river types. For instance, López et al. (2007) constructed an equation to estimate the discharge for coarse-grained rivers. Rupp and Smart (2007) then developed this into an equation for estimating river depth. More recently, machine learning methods have been used to estimate river bathymetry. For instance, Bures et al. (2019) employed a DEM, flow discharge, Manning's n, and support from Random Forrest to model
riverbed topography, while a Deep Neural Network was used by Ghorbanidehno et al. (2021) to map riverbed features from depth-averaged flow speed data.

Neal et al. (2021) categorised four approaches to solve the lack of bathymetric data in flood modelling cases. The first method involves subtracting the estimated river bank-full discharge from the total floodwater to simulate the 'excess discharge' on the floodplains without requiring bathymetric data (Neal et al., 2012). However, it is expected to become inaccurate over large and
complex floodplains (Neal et al., 2012; Sampson et al., 2015). The second method applied the downstream hydraulic geometry (Leopold and Jr, 1953) to estimate riverbed elevation. The relationship between river width, depth, and bank-full discharge used in the method is developed empirically from field observations across many sites (Andreadis et al., 2013; Yamazaki et al., 2013; Gleason and Smith, 2014; Grimaldi et al., 2018). This technique can introduce uncertainties due to various complexities of different rivers into the estimated river (Neal et al., 2021).

The third method applies the Manning's n equation with an assumption of uniform channel over long distances. The formula considers river slope, width, discharge, and friction, to estimate the river depth (Coe et al., 2008-07; Miguez-Macho and Fan, 2012; Brêda et al., 2019). However, real-world rivers are often different from uniform flow conditions, which might cause the flood predictions to be larger or smaller than expected (Neal et al., 2021). In the final method, an observed water surface profile is used to estimate the river bathymetry by applying gradually varied flow equations (Garambois and Monnier, 2015; Brêda
et al., 2019; Andreadis et al., 2020). Despite high accuracy, it is resource-intensive and obtaining the necessary data can be challenging and expensive.

Regardless of any approaches to estimate the river bathymetric data, due to the inability to capture the randomness of the real-world river systems, these estimations still contain errors. These errors can cause the simulated river bathymetries to deviate significantly from the actual ones. Consequently, using these modelled river bathymetries to represent the rivers
in flood inundation modelling can affect the flood predictions. Currently, several studies have investigated the errors in the estimated river bathymetry (Durand et al., 2008; Lee et al., 2018; Moramarco et al., 2019; Kechnit et al., 2024), but they have not considered how these estimations with errors affect the flood model outputs.

For instance, Durand et al. (2008) developed an ensemble-based data assimilation approach for estimating river bathymetry from water surface elevation measurements and the LISFLOOD-FP hydrodynamic model. Using a Monte Carlo-based framework, they also performed a sensitivity analysis to assess how various error sources affected the estimated results. Their study found that errors in some input factors for their approach, such as river roughness and flow conditions, have greater influence than the water surface elevation measurement errors. However, this research did not evaluate how the errors in these river bathymetric estimations can affect flood model outputs with consideration of spatial variability of input factors in the analysis.

Moramarco et al. (2019) introduced a method based on the entropy theory (Shannon, 1948) using channel slope, width, bottom elevation, and a parameter from Alessandrini et al. (2013) to model the river depths. These parameters were estimated using an algorithm that can minimise the observed maximum surface velocity (Moramarco and Singh, 2010). Similar to the Monte Carlo approach, for assessing uncertainties from these parameter estimations, the authors created 1000 river depths from 1000 combinations of parameter values randomly selected from uniform distributions. Kechnit et al. (2024) later extended these techniques to estimate river bathymetry and quantifying uncertainties in larger-scale rivers. Nevertheless, none of these studies investigated how uncertainties in such parameter estimations influence the river depths as well as the flood inundation model outputs, and they have not considered the spatial variability in their analysis.

Lee et al. (2018) introduced a principal component geostatistical method to produce fast bathymetry maps along with the uncertainties. Nevertheless, without using Monte Carlo framework, their research considered uncertainties arising from velocity measurement errors by adding only four Gaussian errors (0.025, 0.01, 0.05, and 0.1 m/s) to the true values without full assessment of the implications of these errors. Hence, their results might not be fully representative for such uncertainties in river bathymetry estimations. Also, their research did not consider how these uncertainties affect the flood inundation model outputs.

Generally, these previous studies have addressed certain gaps in quantifying uncertainties in estimated river bathymetry and show that errors can arise from various sources. However, they have not assessed how the flood inundation model outputs would be affected by errors or uncertainties in the river bathymetry. Additionally, their methods did not consider spatial variability in factors used to estimate river bathymetries and their results are not fully representative.

To fill these gaps, we quantified the uncertainty in flood predictions due to errors in the estimated parameters used in two formulas described in Rupp and Smart (2007) and Neal et al. (2021), and validated by Pearson et al. (2023). Within the Monte Carlo framework, we generated multiple realisations of river bathymetry, then used them to perform a sensitivity analysis to evaluate the impacts of each parameter on flood predictions, individually and collectively. We also considered the spatial variability in the analysis and whether our number of simulations is large enough to represent our results. This work can contribute to studies of other sources of uncertainty to adequately comprehend the uncertainty in flood model outputs. In the next section, we describe a method to explore relationships between the parameters within those two formulas and show a process to examine how errors in these parameters affect the flood predictions.

## 2 Methodology

In this section, we first introduce the study site, necessary data, flood model, and explain the uncertainty propagation process. Next, we define two formulas used for river bathymetry estimation and describe a method to explore the relationships between parameters and river bathymetry from these two equations. We then show how to examine these relationships based on the river of the study site. Finally, we design a sensitivity analysis workflow to quantify the uncertainty in the flood model outputs due to errors in the river bathymetry estimations.

Our data and methodology were based on Nguyen et al. (2025) where the uncertainty in flood predictions due to arbitrary conventions in grid alignment was quantified. To explain, their research is also about how the uncertainty in the process of generating the topographic data like DEM and roughness length can propagate through the flood modelling to the outputs. Hence, their data and methodology can be applied in our research.

Accordingly, we simulated the same flood event using the LISFLOOD-FP flood model and applied a similar method to generate topographic data. Moreover, a Monte Carlo framework was also designed in our research to observe how the uncertainty in estimated river bathymetries propagates through the flood modelling to the outputs. To assess the uncertainty, some similar measurements were used, some were not because they did not provide further information, and some were added to understand better the uncertainty. These similarities will be mentioned in details in the sections below.

### 2.1 Study site and data source

Similar to Nguyen et al. (2025), the Waikanae River, located on the West Coast of the Wellington Region in New Zealand, was used in this paper. Its catchment covers around 149 km$^2$ and spans from the Tararua Ranges to the West Coast. There are recurring flooding issues at this study site that have influenced the regions around the river.

In this study, we simulated a flood event with an 80-year return period that occurred in Waikanae from January $5^{th}$ to $7^{th}$, 2005 and reached its peak on 6th. Here, we focused on fluvial flooding from the Waikanae River. This allowed us to observe how the uncertainty in the estimated river bathymetric data can impact the flood inundation model outputs. Figure 1a depicts our site study extending about 7 km from the Waikanae Water Treatment Plant gauge to the coast. Figure 1b show the flow information recorded at the gauge by the Greater Wellington Regional Council (2005) and the tidal data estimated by the NIWA Tide Forecaster (2005) respectively.

Following the approach of Nguyen et al. (2025), the topographic data - DEM and roughness length - in our paper were generated by an open-source Python package, GeoFabrics (version 0.9.4) developed by Pearson et al. (2023). Specifically, the package sampled and interpolated LiDAR point cloud data downloaded from OpenTopography (2013) onto a 10-metre square grid using Inverse Distance Weighted – an interpolation method has been commonly used in flood modelling (Ibrahim and Fritsch, 2022; Xing et al., 2022; Huang et al., 2023). To represent the river in this process, since the LiDAR only contains the water surface elevations, the estimated riverbed elevation data were then obtained to be included in the point cloud data by subtracting the estimated river bathymetric data or river depths (see Section 2.3) from these water surface elevations. The roughness length was converted to Manning's n using a conversion developed by Smart (2018):

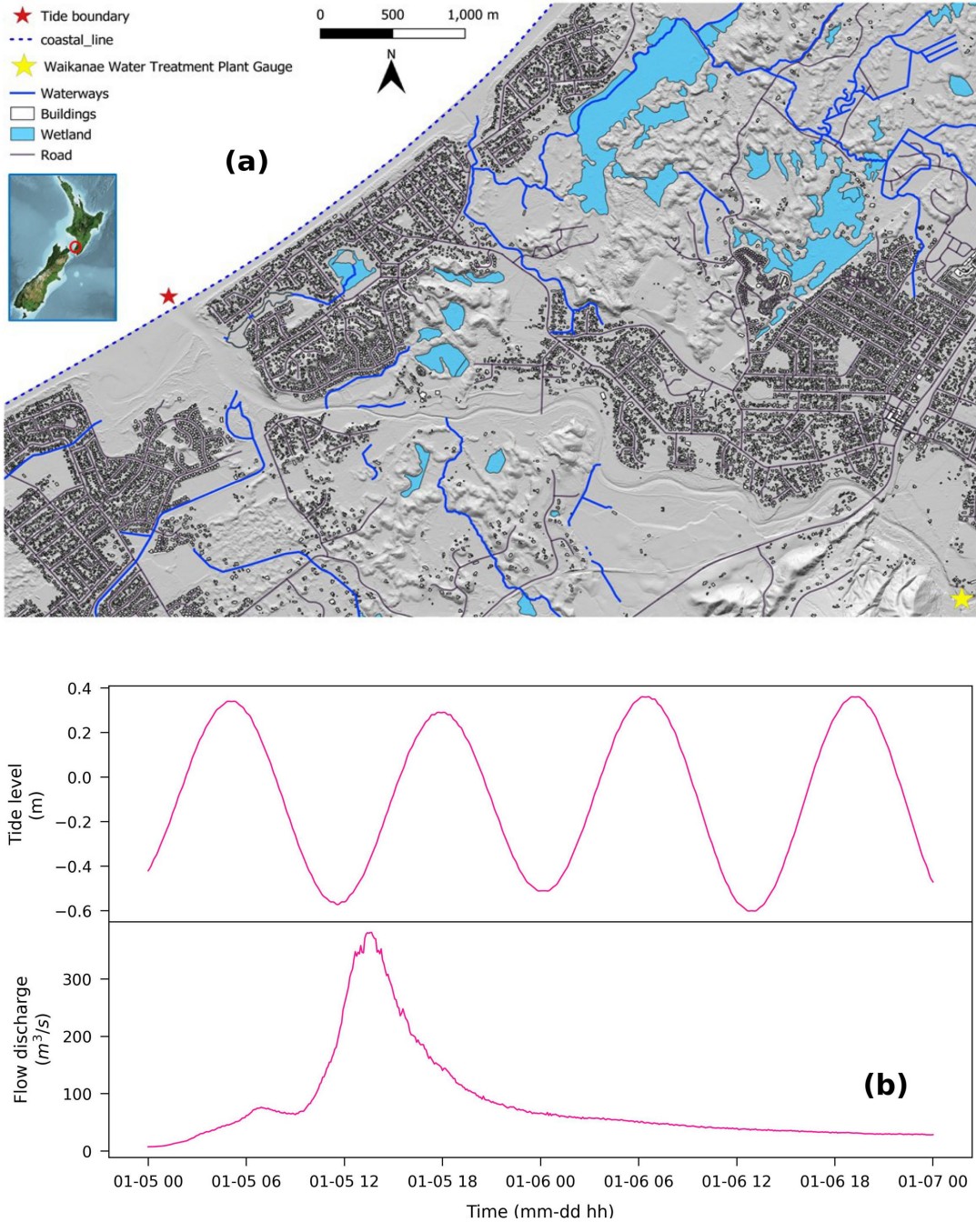

**Figure 1.** Study site and data source (adapted from Nguyen et al., 2025): (a) Waikanae River in Wellington, New Zealand, (b) Waikanae River flow discharge recorded by the (Greater Wellington Regional Council, 2005) and tidal data recorded by the (NIWA Tide Forecaster, 2005) for the flood event from $5^{th}$ to $7^{th}$ January, 2005.

$$n = \frac{kH^{1/6}(\frac{H}{z_o} - 1)}{\sqrt{g}(1 + \frac{H}{z_o}(ln\frac{H}{z_o} - 1))} \tag{1}$$

where $z_o$ is roughness length, $n$ is Manning's n coefficient, $k$ is von Karman's constant (0.41), and $H$ is the flow depth assumed as 1 m in this paper.

## 2.2    Flood model and explanation about uncertainty propagation process

In this study, LISFLOOD-FP (Bates et al., 2010; Neal et al., 2018), a 2D hydrodynamic model, was used to simulate the January-2005 flood event because it is well known for its computational efficiency and highly accurate flood model outputs
(Nguyen et al., 2025). Also, it was calibrated for the Waikanae River in Nguyen et al. (2025). The DEM and Manning's n values, along with the flow information and tidal data mentioned above were used as input into this model.

    In LISFLOOD-FP, the formula to compute the water flow $Qcell$ at the interface index $i + 1/2$, between cells index $i$ and index $i + 1$, over a time step $\Delta t$ is:

$$Qcell_{i+1/2}^{t+\Delta t} = \frac{q_{i+1/2}^t - gh_{flow}^t \Delta t Scell_{i+1/2}^t}{[1 + \frac{g\Delta tn^2|q_{i+1/2}^t|}{(h_{flow}^t)^{7/3}}]} \Delta x \tag{2}$$

where $q^t$ represents the flux at the time $t$, $\Delta x$ denotes the cell width, $Scell$ and $h_{flow}$ are the water surface slope and flow depth between cells (Bates et al., 2010). The flow formula here is displayed for the x direction, the y direction can be obtained analogously. The cell water depth $h_{flow}$ is updated based on the discharge through the four boundaries of that cell as below, where $i$ and $j$ denote the cell coordinates (Shustikova et al., 2019):

$$\frac{\Delta h_{flow}^{i,j}}{\Delta t} = \frac{Qcell_x^{i-1,j} - Qcell_x^{i,j} + Qcell_y^{i,j-1} - Qcell_y^{i,j}}{\Delta x^2}. \tag{3}$$

To further expand on the description of uncertainty propagation through the model given in Section 1, we apply the following chain for easier comprehension:

    Estimated river bathymetric data $\longrightarrow$ riverbed elevations $\longrightarrow$ topographic data (DEM and Manning's n derived from roughness length) generated by riverbed elevations and LiDAR data $\longrightarrow$ inputs to a flood inundation model (LISFLOOD-FP in this study) $\longrightarrow$ affects flood model outputs (extents, depths, etc.)

As indicated in the chain above, the estimated river bathymetric data that contain errors are used to calculate the riverbed elevations (see Section 2.1). These riverbed elevations are then used to represent the river in the topographic data. After that, these topographic data are inputted into the flood model as a discretisation of the floodplain and channel topography to model the water flow. Here, in the flood model, the river as represented in the topographic data controls when, where, and how much the water leaves the channel and starts to flood. Hence, the flood model outputs such as the flood extents and flood depths are
affected by how the river is represented. In the next section, we will describe how the river bathymetric data are estimated.

## 2.3 Method to investigate formulas for river bathymetry estimation

The depths were estimated at regular 10-metre intervals along the river with each point representing an average cross-sectional depth ($h$). From now on, we will use the river bathymetry as an interchangeable term for the river depth. Two formulas were used for this estimation - the Uniform Flow (UF) (Neal et al., 2021) and the Conceptual Multivariate Regression (CMR) (Rupp and Smart, 2007). The CMR formula, designed for coarse-grained rivers, was selected to match with Waikanae River (Gyopari et al., 2014), and the UF formula was chosen for its similar parameters and can be widely applicable. Both are designed in the GeoFabrics and can be presented through a general equation as below:

$$h = (\frac{nQ}{wS^{\beta}})^{\frac{1}{1+\alpha}}. \tag{4}$$

The cross-section width ($w$) at bank-full river and river slope ($S$) were estimated from LiDAR data as detailed in Pearson et al. (2023). The river bank-full flow ($Q$) and river Manning's ($n$) were obtained from Henderson and Collins (2018) at NIWA. For the $\alpha$ and $\beta$ coefficients, the UF formula used constant values of 2/3 and 1/2 respectively, while the CMR formula, designed for coarse-grained rivers, applied 0.745 and 0.305 respectively with a constant value of 0.162 for Manning's n. The exponents and value ranges of each parameter are shown in Table 1 and some of them are explained in Appendix A. In our research, we assumed the errors in the estimated river bathymetries arising from the errors in these parameters owing to estimation. Due to time-consuming and complex nature of processing simulations for river Manning's n and coefficients $\alpha$ and $\beta$, we focused solely on the errors in the river slope, bank-full flow, and width in this paper.

| Parameters | Slope (S) | Flow (Q) | Width (w) | Manning's n (n) |
|---|---|---|---|---|
| Exponents - CMR | 0.175 | 0.573 | 0.573 | 0.573 |
| Exponents - UF | 0.3 | 0.6 | 0.6 | 0.6 |
| Minimum values along the river | 0.3 (m/km) | 145.2 (cumec) | 19.4 (m) | 0.162 (for CMR) & 0.0377 (for UF) |
| Maximum values along the river | 7.2 (m/km) | 146.2 (cumec) | 99.6 (m) | 0.162 (for CMR) & 0.0436 (for UF) |
| Mean values along the river | 4.0 (m/km) | 146.1 (cumec) | 35.6 (m) | 0.162 (for CMR) & 0.0433 (for UF) |

**Table 1.** The exponents of parameters in the Conceptual Multivariate Regression and Uniform Flow formulas (see Appendix A), and the value ranges (minimum, maximum, and mean) of parameters along the Waikanae River – the river slope, bank-full flow, width, and Manning's n – used to explore their relationships with the river bathymetry in both formulas.

Before Monte Carlo simulation process, we explore the relationship between these parameters and the river bathymetries estimated by the UF and CMR formulas. At first, the mean value over the entire river section of each parameter is calculated as seen in Table 1. We then increase the mean value of each parameter, except for the river Manning's n, from 50% to 200% while keeping other parameters constant. This method allows us to observe how the river bathymetries from the two formulas are affected when a parameter is varied. The result analysis of this part is mentioned in Section 3.1.

## 2.4 Method to evaluate relationships of bathymetry and parameters in the formulas on the study site

We then make the findings of Section 2.3 more concrete by investigating how they play out along the Waikanae River. We look at the best estimates of the parameters (slope, bank-full flow, and width) and the Waikanae River bathymetry along with the Monte Carlo simulations of their variances. Here, we examine how each parameter and their combination along with errors are correlated with the river bathymetry. Specifically, we visualise the variation of each parameter along the river and the resulting variation in river bathymetry. We plot both the values as a function of location along the river and scatter plots of their relationships. We also plot the along-river bathymetries for the combined errors of all three parameters. Three scatter plots depict the relationships between the variance of each parameter and these combined river bathymetries. All of these visualisations and analysis are provided in Section 3.2. In the next section, we detail how to generate these simulated parameters and corresponding river bathymetries and examine their variations on flood predictions.

## 2.5 Monte Carlo simulation process

Figure 2 shows a Monte Carlo simulation process undertaken in this study. To describe the framework in this figure, we divide this section into two subsections. The first is about simulation process and the second is about statistical analysis.

### 2.5.1 Simulation process

At first, to generate multiple simulated parameters with the same amount of errors, we used GeoFabrics package to gather their best estimates along the river derived from LiDAR (river slope and width) and estimated by NIWA (river bank-full flow). Due to no information about the sources of errors, we assumed that their expected errors would be unbiased and normally distributed with zero mean and a standard deviation of 10% of the best-estimated values. This 10% was chosen because: (i) many observed cross-sectional riverbed elevations are within the simulated ensemble range (min-max) of simulated riverbed elevations - calculated from the simulated river bathymetric data (described in detail later in this section) - as seen in Fig. 3; and (ii) with the same amount of errors, we can then compare the influences of those errors, between datasets, on the flood model outputs. Although we do not know what the true errors are in these parameters, these assumed but reasonable ones from the Monte Carlo framework can still meaningfully indicate how the estimated bathymetric data can affect the flood model outputs. In future research, the measured errors can apply the framework already built in this study to compare and confirm the results.

Within the process of generating the simulated errors for each parameter, we spatially model the variation of the errors along the river with a Gaussian variogram. This was implemented using Gstat, an open-sourced R package developed by Pebesma (2004); Gräler et al. (2016). The Gaussian variogram was chosen because it smoothly represents how errors might vary over space, ensuring that points closer to each other along the river have more similar errors. This is particularly suitable for river slope, bank-full flow, and width, which tend to change gradually rather than abruptly along the river.

Next, this variogram was employed to generate 50 unconditional simulations of these errors from a normal distribution with a zero mean and a standard deviation equal to the expected error. Each simulation is different along the river but with the same characteristics of the variogram. 50 realizations of each parameter were then generated by adding these simulated errors to the

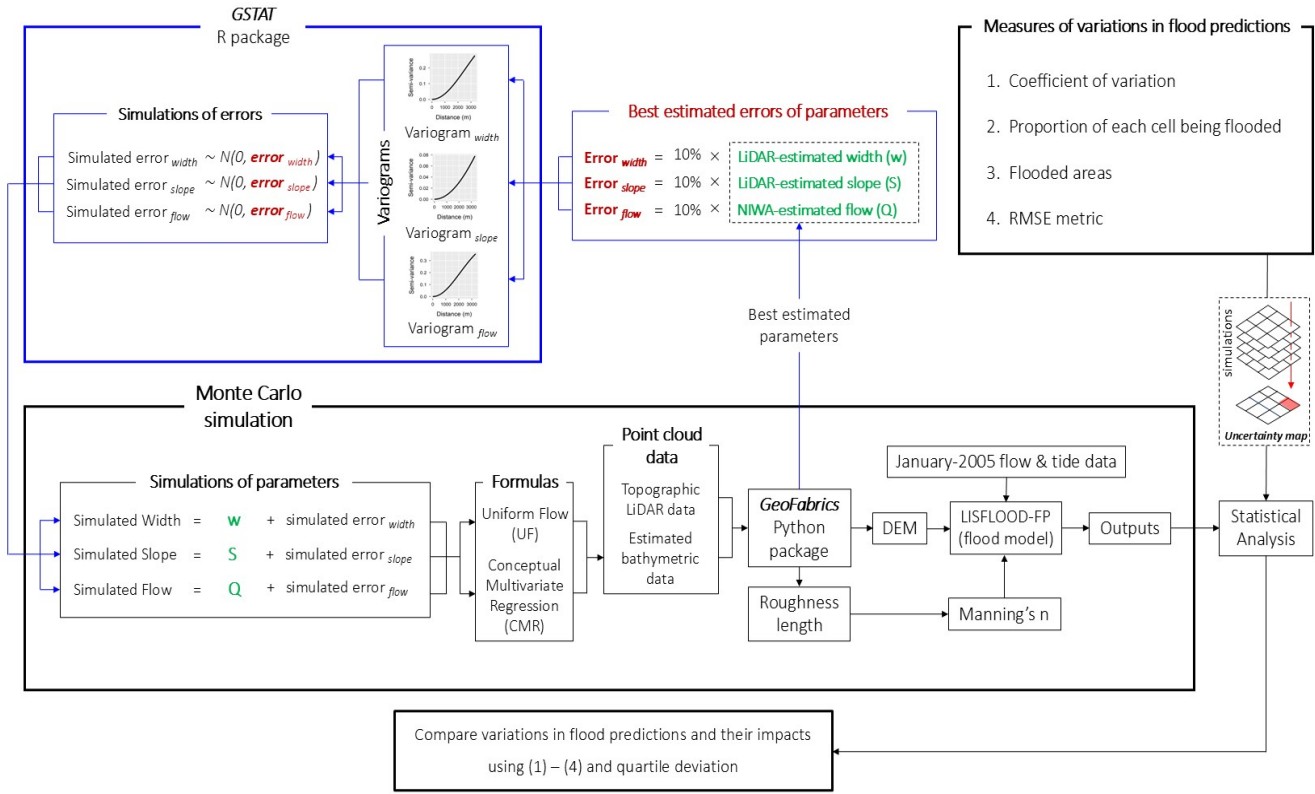

**Figure 2.** Process to quantify uncertainty in flood predictions using river bathymetries estimated by the Conceptual Multivariate Regression and Uniform Flow formulas, with associated error distributions in parameters: river slope, bank-full flow, and width.

corresponding best estimated parameter. This quantity was chosen for time efficiency as testing with a larger number we found
that using more representations did not considerably impact the results.

Here, we selected the normal distribution because we assumed our parameters from the estimations and measurements provide the most accurate values, making them the most probable. Moving further from them, the probability of errors should decrease. Additionally, we presumed the errors can be both negative and positive, balanced around zero. Besides, the lack of information about the true errors led us to use unconditional simulation. This method provides a wide range of errors to
understand better their relationships with the river bathymetries and how they impact flood predictions.

Subsequently, 400 realizations (eight datasets of 50 simulations) of the river bathymetries were created in total: 50 representing the variation in each of the three parameters (river slope, bank-full flow, and width) and additional 50 using the combined variations in the three parameters times the two formulas (UF and CMR) used for calculation. Next, these simulated river bathymetries were then subtracted from the LiDAR-estimated water surface elevations to obtain the
simulations of riverbed elevations. Eight datasets of these simulated river data were organised and presented in the Table 2.

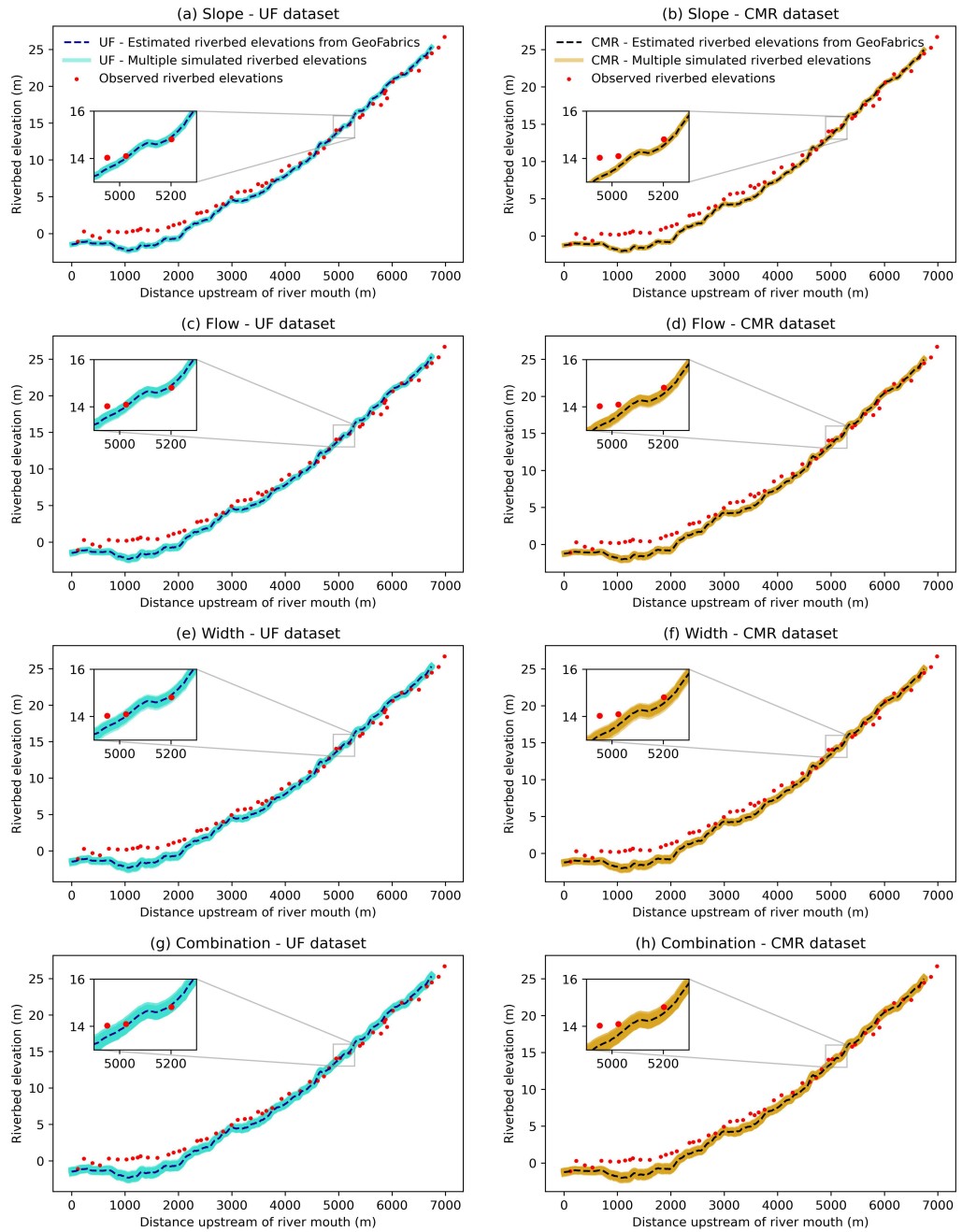

**Figure 3.** Observed cross-sectional, best estimated (from GeoFabrics of Pearson et al. (2023)), and simulated riverbed elevations at the Waikanae River. The best estimates and simulations of riverbed elevations computed using the Uniform Flow formula are in the first column: (a) slope, (c) bank-full flow, (e) width, and (g) combined. The ones calculated using the Conceptual Multivariate Regression formula are in the second column: (b) slope, (d) bank-full flow, (f) width, and (h) combined. The color shading represents multiple simulated riverbed elevations (span of simulations).

| No. | Dataset names (Parameter - formula) | Descriptions |
|---|---|---|
| 1 | Slope - UF | 50 simulated river depths/riverbed elevations estimated by the Uniform Flow formula when the simulated errors selected from N(0, error$_{slope}$) were added into the LiDAR-estimated slope |
| 2 | Flow - UF | 50 simulated river depths/riverbed elevations estimated by the Uniform Flow formula when the simulated errors selected from N(0, error$_{flow}$) were added into the NIWA-estimated flow |
| 3 | Width - UF | 50 simulated river depths/riverbed elevations estimated by the Uniform Flow formula when the simulated errors selected from N(0, error$_{width}$) were added into the LiDAR-estimated width |
| 4 | Combination - UF | 50 simulated river depths/riverbed elevations estimated by the Uniform Flow formula when the simulated errors selected from N(0, error$_{slope}$), N(0, error$_{flow}$), and N(0, error$_{width}$) were simultaneously added into the LiDAR-estimated slope, NIWA-estimated flow, and LiDAR-estimated width respectively |
| 5 | Slope - CMR | 50 simulated river depths/riverbed elevations estimated by the Conceptual Multivariate Regression formula when the simulated errors selected from N(0, error$_{slope}$) were added into the LiDAR-estimated slope |
| 6 | Flow - CMR | 50 simulated river depths/riverbed elevations estimated by the Conceptual Multivariate Regression formula when the simulated errors selected from N(0, error$_{flow}$) were added into the NIWA-estimated flow |
| 7 | Width - CMR | 50 simulated river depths/riverbed elevations estimated by the Conceptual Multivariate Regression formula when the simulated errors selected from N(0, error$_{width}$) were added into the LiDAR-estimated width |
| 8 | Combination - CMR | 50 simulated river depths/riverbed elevations estimated by the Conceptual Multivariate Regression formula when the simulated errors selected from N(0, error$_{slope}$), N(0, error$_{flow}$), and N(0, error$_{width}$) were simultaneously added into the LiDAR-estimated slope, NIWA-estimated flow, and LiDAR-estimated width respectively |

**Table 2.** Dataset descriptions of simulated river bathymetries estimated by the Uniform Flow and Conceptual Multivariate Regression formulas with errors in parameters: river slope, bank-full flow, and width.

Similar to Durand et al. (2008); Moramarco et al. (2019); Kechnit et al. (2024), and especially Nguyen et al. (2025), our research also applied a Monte Carlo framework to generate 50 DEMs and 50 Manning's n maps from those 50 simulated riverbed elevations and LiDAR data from OpenTopography (2013) using the method described in Section 2.1 for each dataset. These 50 DEMs and 50 Manning's n maps are the same except for the river locations due to the use of 50 different simulated riverbed elevations. Hence, we only focus on analysing the variation in the simulated river bathymetric data used to generate these riverbed elevations instead of those simulated topographic data (see Section 3.2). The DEMs and Manning's n maps that include the simulated river bathymetric data, along with the January-2005 flow and tidal data mentioned in Section 2.1, were then used in the LISFLOOD-FP flood model to produce 50 maximum water depths (MWDs) and 50 maximum water surface elevations (MWSEs) for further statistical analysis.

### 2.5.2 Statistical analysis

To assess the uncertainty in flood predictions, we measured the variability in these simulated MWDs by computing their mean (mMWDs) and standard deviation (sdMWDs) to calculate the coefficient of variation (covMWDs). The proportion of simulations in which a given pixel was flooded (pFs) were also computed to distinguish where was always flooded, never flooded, and sometimes flooded throughout these realizations. However, different to Nguyen et al. (2025), mMWDs and sdMWDs were not considered in the research due to no useful information.

The covMWDs and pFs were then mapped with probability density functions for each set. Here, pixels with mMWDs of 0.1 m or greater were classified as flooded and included in the analysis, while those of shallower than 0.1 m were excluded. Apart from that, the oceanic zone and river were also removed to focus on the variations in the floodplains. Additionally, we computed expected flooded area or expected flood extent, a metric often employed by decision-makers, for each simulation for comparison. The expected flood extents were calculated based on these pFs by multiplying the area of one pixel (10 m x 10 m) with number of pixels that were always and sometimes flooded.

To examine variations in flood predictions of the eight datasets, side-by-side boxplots were applied to visualise the distributions of flood extents and those of covMWDs. We compared the magnitude of their variations using the quartile deviation metric which was also employed by Nguyen et al. (2025). In our research, we went further than Nguyen et al. (2025) by validating each flood simulation - MWSE with the observed data. Due to the lack of a thorough map of measured flood levels or satellite-based water surface elevations, we used the observed flood levels under point format provided by Wallace (2010). The Root Mean Square Error (RMSE) metric was harnessed for these validations. Locations of the observed data where the flood model predicted to be dry across all the simulations were removed to ensure the RMSE focuses only on predicted flooded regions and to avoid skewing the RMSE. We then visualised the distribution of RMSEs across simulations through side-by-side boxplot for comparison.

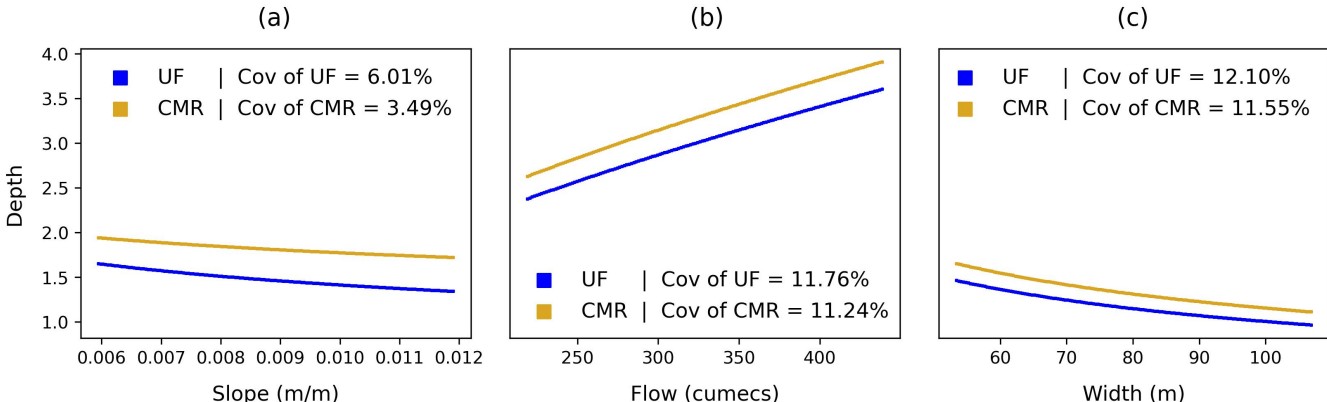

**Figure 4.** Formula investigation: relationships between (a) river slope, (b) bank-full flow, and (c) width with river bathymetries estimated by the Uniform Flow and Conceptual Multivariate Regression formulas. Each plot shows how the river bathymetries are correlated with increasing the mean value of a parameter (see Table 1) from 50 % to 200 % while keeping others constant.

## 3   Results

In this section, we showcase the findings from investigating the relationships between the bathymetry and parameters in two formulas, followed by the results from evaluating how these findings playing out with the study site, the Waikanae River. We then illustrate the results from comparing variations in flood predictions across eight datasets.

### 3.1   Findings from investigating formulas for river bathymetry estimation

Figure 4 shows the relationships between the river bathymetry and three parameters – river slope, bank-full flow, and width. As seen in Fig. 4a, a steeper river is prone to be shallower. This inverse relationship is mathematically represented in both formulas, where the slope appears in the denominator. Physically, when the river width and flow do not vary, and the sediment effects are not considered, it is expected that in steeper sections, the water tends to flow faster and spend less time interacting
with the riverbed. Therefore, its force has a smaller impact on the river bathymetry.

Figure 4b shows that a deeper river tends to have larger flow. This proportional relationship can be explained mathematically in both formulas, where the flow is in the numerator. Physically, it can be understood that, in the river sections where the river width and slope do not vary, and the sediment influences are not considered, the increased flow has greater water force, which is correlated with a higher impact on the river bathymetry than smaller flow.

Figure 4c depicts that wider river is likely to be shallower. It can be comprehended that in wider river sections, where the river slope is unchanged, the constant water volume spreads out and reduces its force which has a smaller impact on the river bathymetry. This inverse relationship is also presented in both formulas, where the width is positioned in the denominator.

Based on the coefficients of variation, the variations in the river bathymetries are more strongly correlated with the variations in the width than the flow, and much more than the slope. Physically, the width can control the water distribution on the riverbed,

which is strongly connected to the impact magnitude of the water force on the river bathymetry. Meanwhile, although higher flow increases the water force, it does not control water distribution as effectively as the width does, which has less correlation with the river bathymetry. The slope is primarily associated with the flow velocity rather than the water distribution, so its changes are much less correlated with the changes in the river bathymetry.

Mathematically, the width and flow have the same higher exponents in both formulas (0.6 for UF, 0.573 for CMR) compared to the slope (0.3 for UF, 0.175 for CMR). This highlights why variations in the width and flow have stronger correlations with the river bathymetry. Additionally, with width in the denominator and flow in the numerator, flow variability is slightly less correlated with the variability in the river bathymetry than width variability.

Apart from that, in Fig. 4, the river bathymetry estimated by the CMR formula is generally greater than that estimated by the UF formula. This difference arises from many factors, but mainly from the friction and its exponent in this case. The mean friction along the river used for the UF formula (0.0432) is lower than the constant friction (0.162) in the CMR formula. Also, its exponent in the UF formula (0.6) is higher than in the CMR formula (0.573). However, when considering other factors, for instance, if the width continues to increase and the slope or flow decreases, the river bathymetries estimated by both formulas can converge, switch positions, and then diverge again. Besides, based on the CoVs, for the same amount of variation in the parameters, the river bathymetries estimated by the UF formula have higher variability than that estimated by the CMR formula. To explain, the exponents of the slope (0.3), flow (0.6), and width (0.6) in the UF formula are higher than in the CMR formula (0.175, 0.573, 0.573, respectively).

Overall the variation in the river width corresponds to the largest variability in the river bathymetry followed by variations in the river flow and slope. Besides, in this case, the UF formula generates shallower river than the CMR formula, mainly due to the differences in the friction and its exponent. However, the situation can change depending on how other factors such as the slope, width, and flow alter. Additionally, the UF river bathymetry is more sensitive to variations in these parameters than the CMR river bathymetry.

The above findings are based on the variation in the river bathymetry when a parameter is changed while others remain constant. Also, we have not considered other factors such as sediment load in this analysis. Hence, these results should not be used to fully reflect the real-world river systems. In the next section, we will observe the changes of these parameters and their simulations along the Waikanae River and how they are correlated with the changes in the corresponding river bathymetries to see if the results match with the findings in this section.

## 3.2   Findings from evaluating relationships of bathymetry and parameters in the formulas on the study site

In this section, we first analyse Fig. 5 by dividing the distance between Waikanae River Treatment Plant gauge (upstream) and the coast into two parts - from the river upstream to 1000 m downstream (upstream reach) and from 1000 m downstream to the coast (downstream reach). Based on this, we focus on analysing the upstream reach of the slope (first row of Fig. 5), flow (second row), width (third row), and combined (forth and fifth rows) datasets. We then compare the two formulas in the upstream reach and then in the downstream reach. After this, Fig. 6 will be examined.

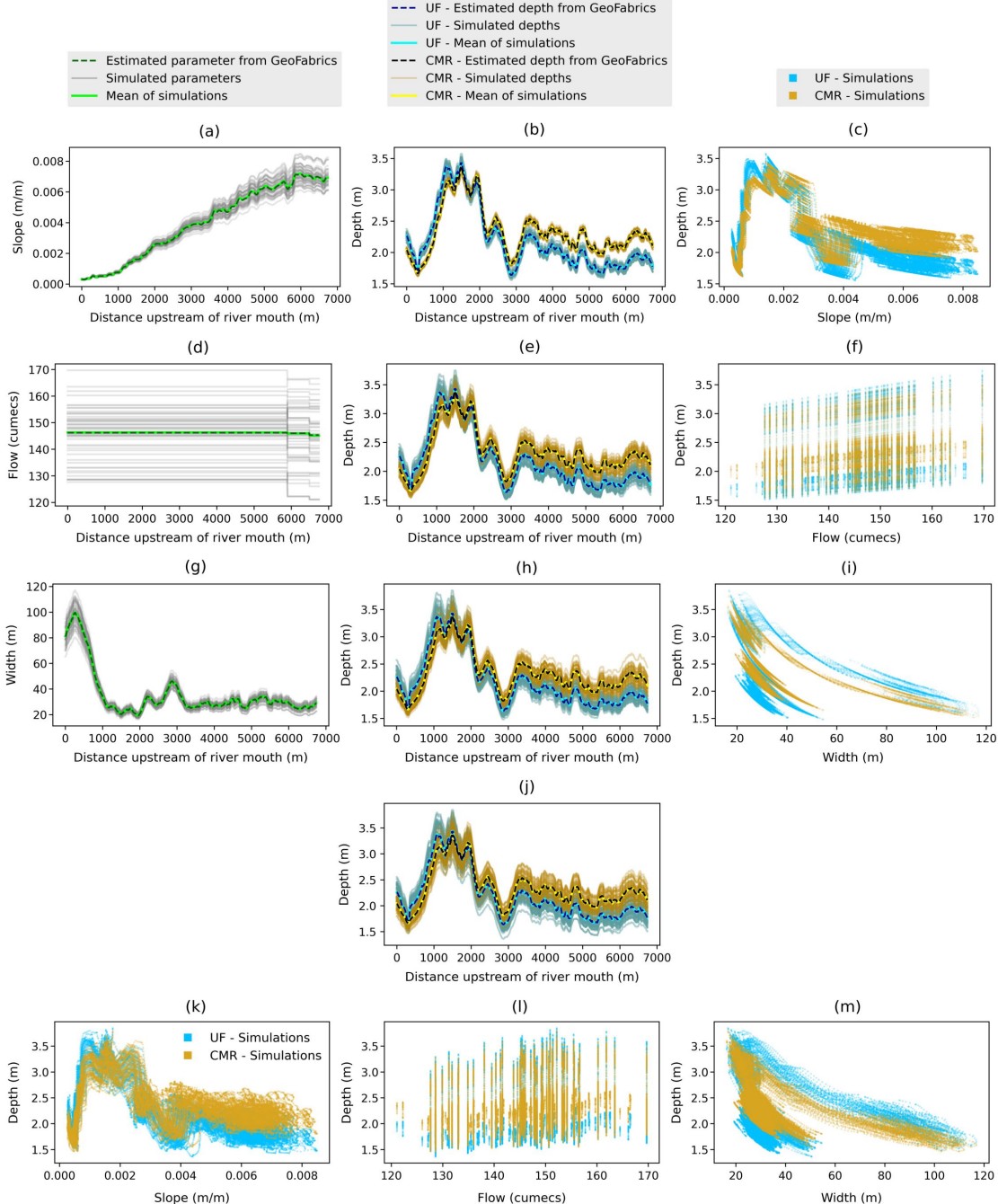

**Figure 5.** Relationships between simulated Waikanae River slopes, bank-full flows, widths, and bathymetries estimated by the Conceptual Multivariate Regression and Uniform Flow formulas. First row: (a) simulated slopes, (b) corresponding bathymetries, and (c) their relationships. Second row: (d) simulated bank-full flows, (e) corresponding bathymetries, and (f) their relationships. Third row: (g) simulated widths, (h) corresponding bathymetries, and (i) their relationships. Fourth and fifth rows: (j) simulated bathymetries using all simulated parameters, and their relationships with simulated (k) slopes, (l) bank-full flows, and (m) widths.

For the slope dataset, Fig. 5a-b indicate that, in the upstream reach, the Waikanae river becomes gentler when it also deepens. In this case, despite variability of other parameters (i.e. river width and flow) along the river, the relationship between slope and bathymetry still aligns with findings in Section 3.1. Their simulations also follow this trend as seen in Fig. 5c.

For the flow dataset, in the upstream reach of Fig. 5d-e, when the Waikanae River becomes deeper, its flow shows only a slight increase, from 145.196 to 146.194 cumecs, with the highest value remaining constant for the next 6000 m downstream. This implies that the bathymetry along this river is not strongly correlated with the bank-full flow. However, in Fig. 5f, the simulated rivers slightly deepen when the simulated flow increases. This pattern is still consistent with observations from Section 3.1, even though other simulated parameters (i.e. river width and slope) vary along the river.

For the width dataset, in Fig. 5g-h, in the upstream reach, the Waikanae River width resembles a reversed version of its bathymetry, showing an inverse relationship. In this situation, in spite of variations of other parameters (i.e. river slope and flow), the relationship of the river width and bathymetry still follows the results found in Section 3.1. Their simulations also indicate this trend in Fig. 5i.

For the combined dataset, Fig. 5j-m show the same patterns as what we found above when analysing each parameter dataset. Specifically, in the upstream reach, the simulated bathymetries and bank-full flows are not strongly correlated with each other. Apart from that, the simulated river slopes decrease as the simulated bathymetries increase. Finally, the shapes of the simulated river widths are reserved versions of the simulated river bathymetries, showing their inverse relationship.

Between two formulas, in the upstream reach of Fig. 5b, e, h, and j, the river bathymetries estimated by the UF are lower than the CMR formula mainly due to the difference in the friction and its exponent, as explained in Section 3.1. However, in the downstream reach, both formulas generate shallower rivers in which the UF bathymetries are greater than the CMR bathymetries. This is where the river slope decreases 80 % from about 0.001 m/m to about 0.0002 m/m. Simultaneously, its width increases up to 400 % from approximately 20 m to around 100 m.

Furthermore, in the downstream reach, given the flat terrain, the increase in width outweighs the decrease in slope. Mathematically, the slope and width are in the denominator of both formulas, indicating their inverse relationships with the river bathymetries. Moreover, the slope drop (within 80 %) and its exponents (0.3 and 0.175 for the UF and CMR formulas) are much smaller than the width increase (within 400 %) and its exponents (0.6 and 0.573 for the UF and CMR formulas). Consequently, the river bathymetries are affected by the increase in the river width than the decrease in the slope. Besides, as mentioned in Section 3.1, when the width starts increasing and the slope keeps decreasing, the river bathymetries of both formulas first converge, then diverge, with the UF bathymetries eventually exceeding the CMR bathymetries.

Figure 6 shows the variations in the simulated bathymetries across the spatial domain when the associated error distributions with the same percentage standard deviation were added into the slope, flow, and width both individually and simultaneously. In both formulas, the ranges of coefficients of variations increases between the slope, flow, and width datasets. It indicates that the variation in the river width is associated with the largest variability in the river bathymetries, followed by the river flow and slope. Moreover, the colours of the UF-formula river bathymetries are darker than those of the CMR-formula ones. This demonstrates the UF-formula bathymetries exhibit larger variability than those from the CMR formula. These all results are consistent with the findings in Section 3.1.

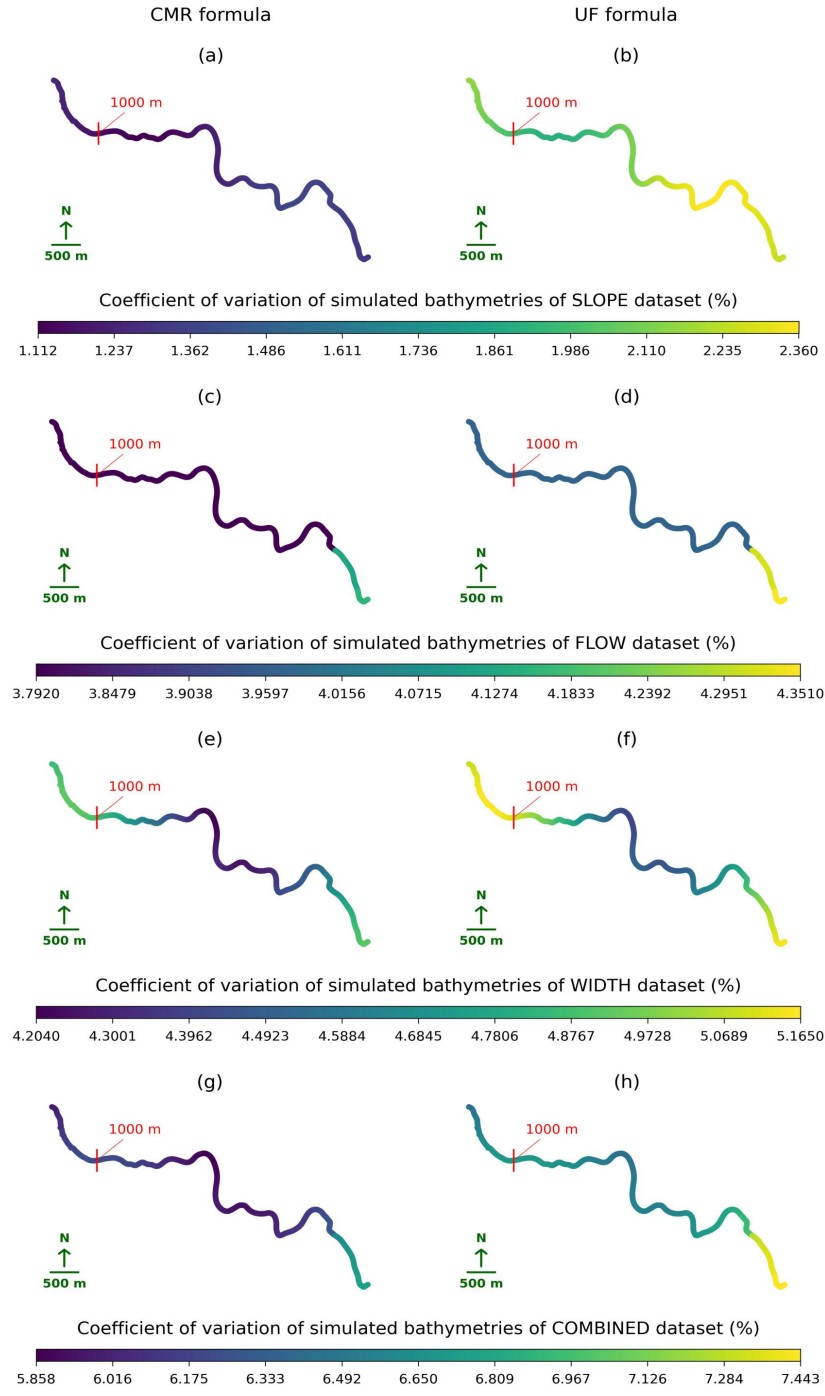

**Figure 6.** Variations in the simulated Waikanae River bathymetries due to associated error distributions in parameters: the Conceptual Multivariate Regression formula - (a) slope, (c) bank-full flow, (e) width, and (g) combined; the Uniform Flow formula - (b) slope, (d) bank-full flow, (f) width, and (h) combined.

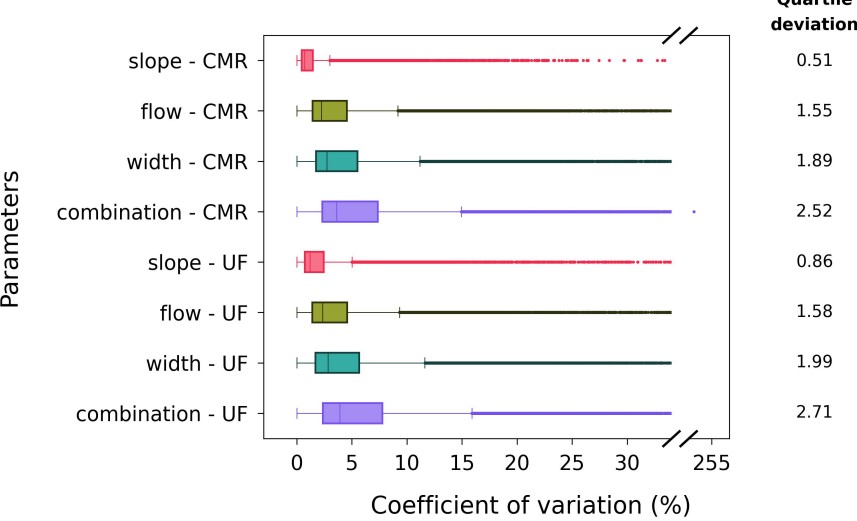

**Figure 7.** Distributions of coefficients of variations of maximum water depths (covMWDs) for eight datasets (slope-, flow-, width-, and combination-CMR and -UF datasets).

Overall, in the upstream reach, despite a slight rise in the simulated bathymetries when the simulated flow increase, they are not strongly correlated with each other. Along this distance, the simulated river becomes deeper as their slopes become gentler. The simulated river widths are opposite versions of the simulated bathymetries, demonstrating their inverse relationship. Despite the simultaneous variability along the river of these parameters, their relationships with the bathymetry are still consistent with the findings in Section 3.1. In the downstream reach, the river becomes shallower when it widens with a mild drop in the slope. Besides, for both formulas, the variation in the river width corresponds to the largest variability in the river bathymetries, followed by the river flow and slope. The UF-formula river bathymetries have more variations than the CMR-formula ones across three parameters. In the next section, we investigated how these variations in the bathymetries affect the flood predictions.

### 3.3 Comparison of variations in maximum water depths

The variations in covMWDs using the eight datasets are presented in Fig. 7 with their quartile deviations for comparison. We observed that, in both formulas, the slope dataset exhibits the smallest variability in the covMWDs, followed by the flow, width, and combination datasets. Furthermore, across all parameters, the UF-formula datasets have higher variability in the covMWDs than the CMR-formula datasets.

The order of variation magnitudes between datasets is visible in Fig. 8, especially in the green zoomed in images. Specifically, in both formulas, more locations with covMWDs less than 1.5 % are found in the slope dataset than in the flow, followed by the width, and then the combination datasets. The covMWDs larger than 1.5 % are mainly observed at the edges of the flood

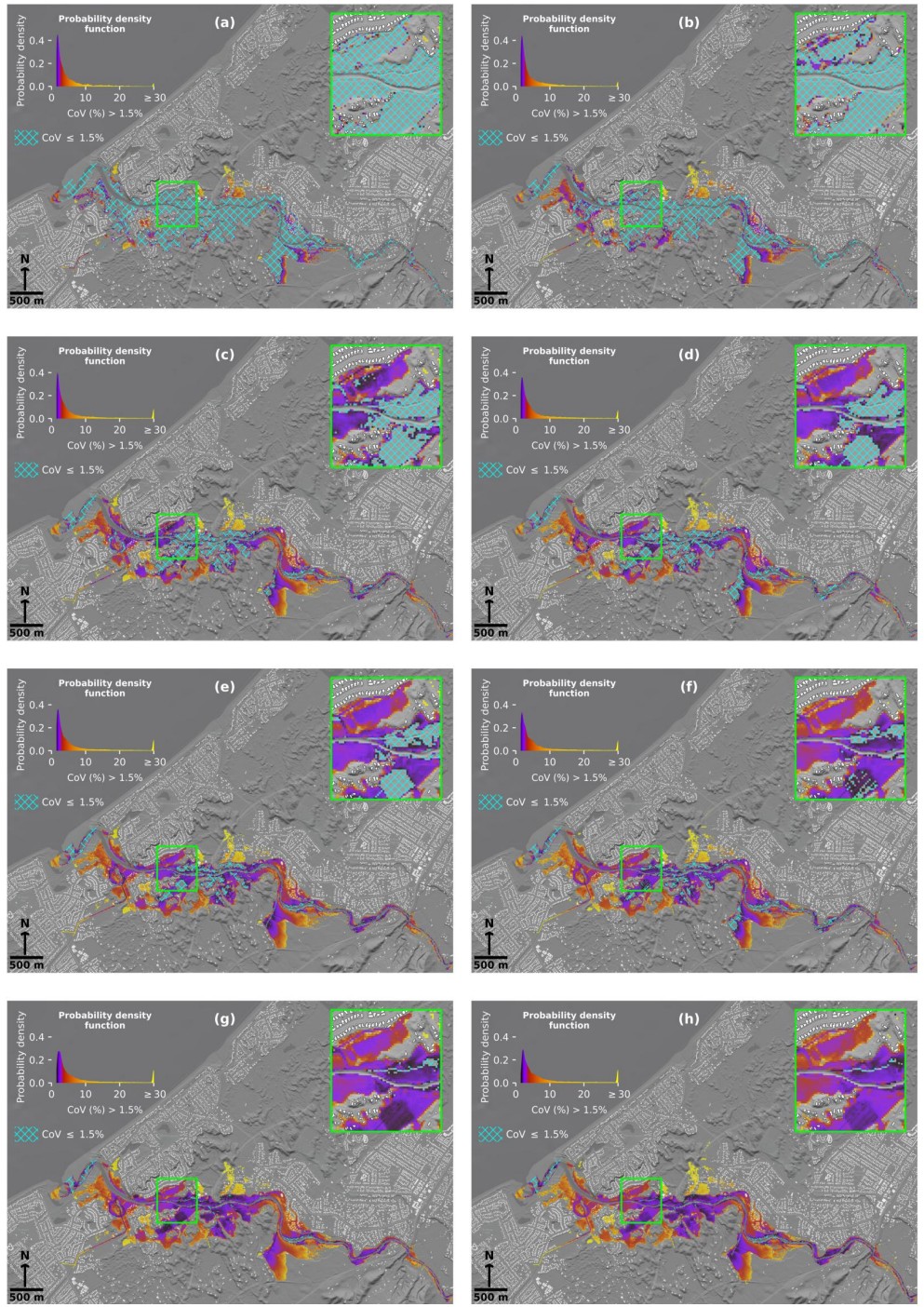

**Figure 8.** Variations in January-2005 maximum flood depths based on simulated Waikanae River bathymetries estimated by parameters with associated error distributions: the Conceptual Multivariate Regression formula - (a) slope, (c) bank-full flow, (e) width, and (g) combined; the Uniform Flow formula - (b) slope, (d) bank-full flow, (f) width, and (h) combined.

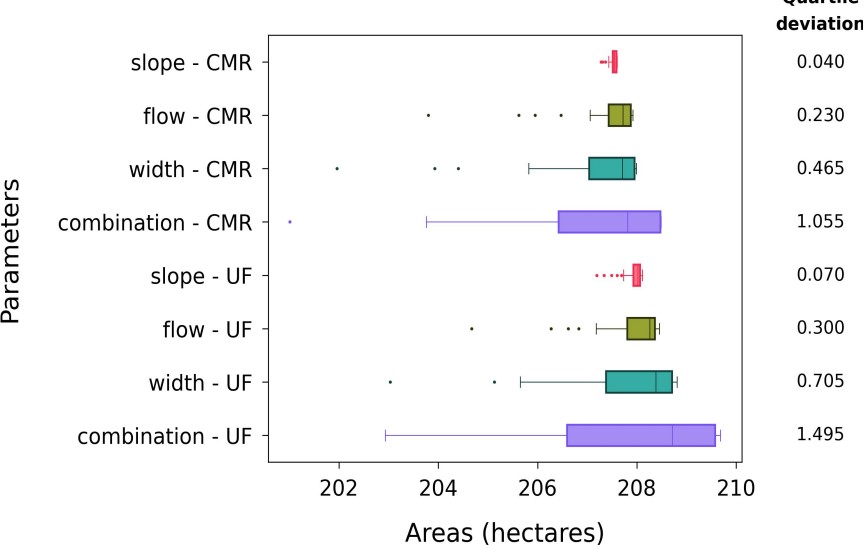

**Figure 9.** Distributions of flood extent for eight datasets (slope-, flow-, width-, and combination-CMR and -UF datasets).

extents around midstream and tend to decrease closer to the river. It is also clear in these figures that the datasets using the CMR formula have more locations with covMWDs below 1.5 % than those using the UF formula. For the combination dataset, in the green zoomed-in image, we can see that the colours of covMWDs of the CMR formula are darker than those of the UF formula. These orders of variation magnitudes in flood depths between datasets follow those in the river bathymetries found in Section 3.1 and 3.2.

To explain, between parameter datasets, the small variability in the river bathymetry corresponding with the variation in the river slope does not significantly affect the water spreading into the floodplain, unlike the variations in the river bank-full flow and width. The impacts of all these variations become more apparent in floodplains farther from the river, especially at flood boundaries in midstream, where the water has less direct connection with the river. Between two formulas, because the variations in the UF-formula river bathymetries are higher than the CMR-formula ones as seen in Fig. 6, the variations in the

flood depths of the UF-formula datasets are also higher than the CMR-formula datasets.

## 3.4 Comparison of variations in flood extents

Figure 9 shows a comparison of flood extents between the eight datasets. In both formulas, the slope datasets have the smallest variability in the flood extent followed by the flow, width, and the combination datasets. The order of these variation magnitudes in flood extent between datasets align with those in the river bathymetries, as noted in Section 3.1 and 3.2. The blue zoomed-in

images in Fig. 10 can visualise these flood extent differences.

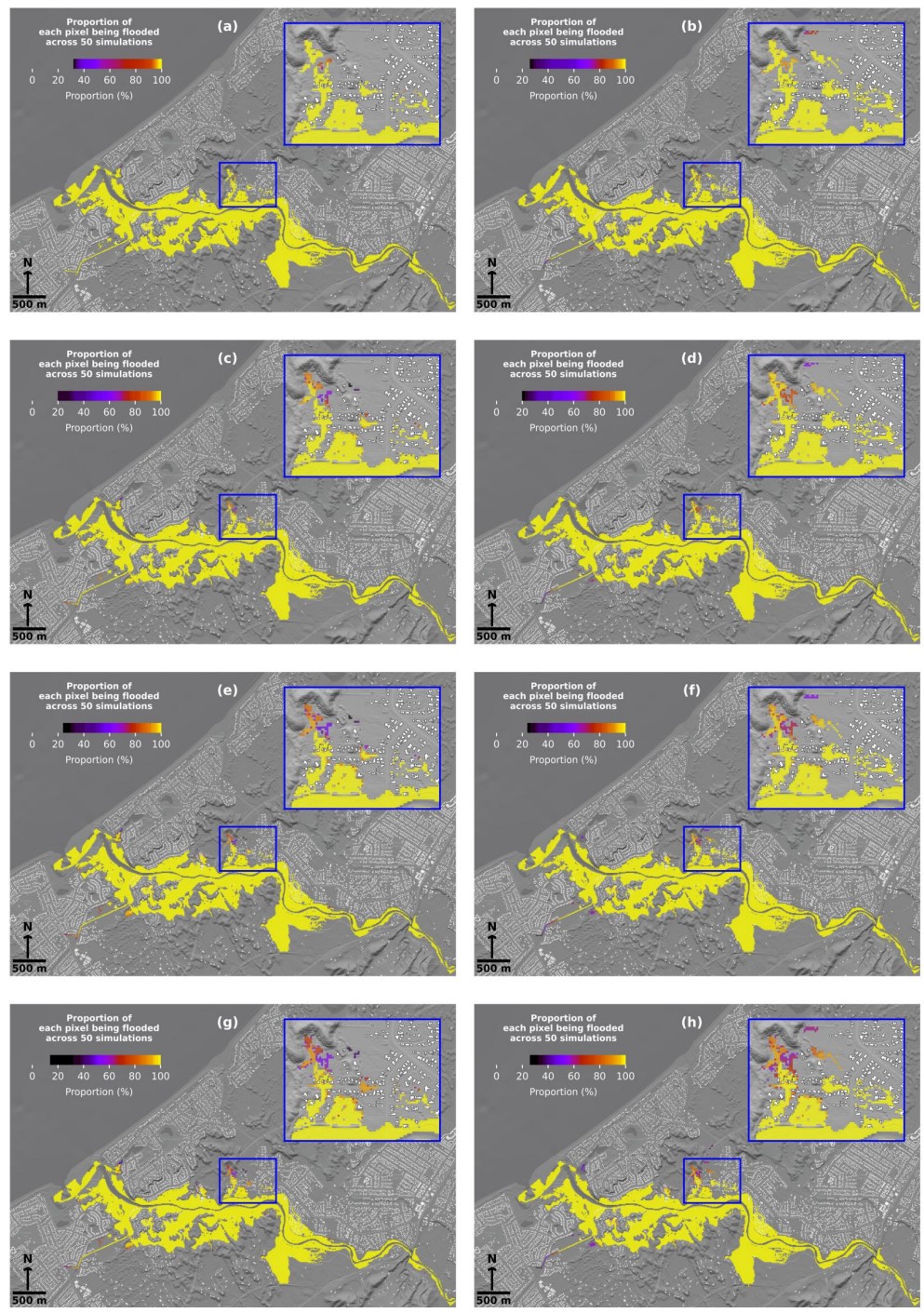

**Figure 10.** Variations in January-2005 flood extents based on simulated Waikanae River bathymetries estimated by parameters with associated error distributions: the Conceptual Multivariate Regression formula - (a) slope, (c) bank-full flow, (e) width, and (g) combined; the Uniform Flow formula - (b) slope, (d) bank-full flow, (f) width, and (h) combined.

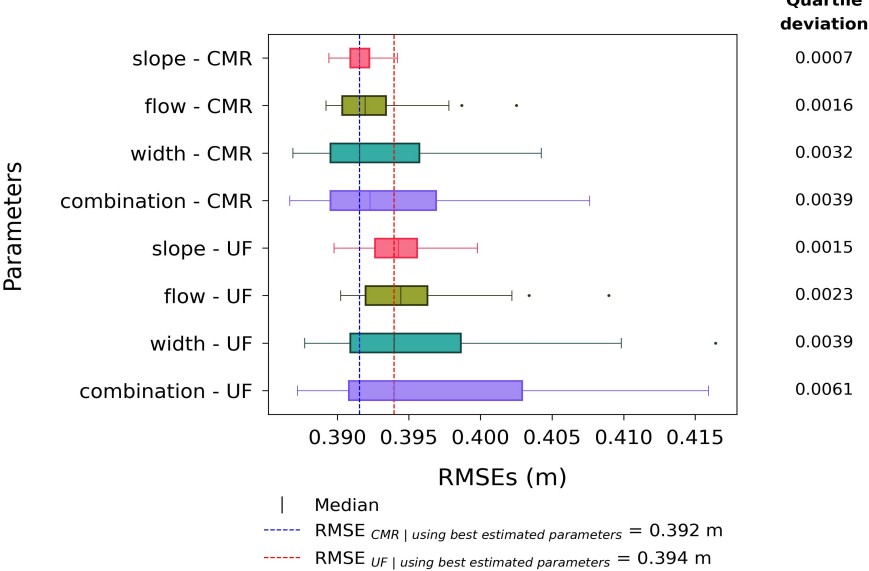

**Figure 11.** RMSE distributions for predicted flood levels of eight datasets (slope-, flow-, width-, and combination-CMR and -UF datasets) compared to the January-2005 observed flood levels. The RMSEs were calculated using the method described in Section 2.5.2

The locations flooded less than 100 % of the time increase between the slope, flow, width, and combination datasets as seen in the blue zoomed-in images of Fig. 10. To explain, the bathymetry determines the water volume the river can hold, which influences how much water can exceed the riverbank and extend in the floodplain. Hence, with the same amount of flood water from the January-2005 event, a small variation in the river bathymetry can result in a small variation in the flood extent and vice versa. This demonstrates the order of variation magnitudes in flood extents between datasets follows that in the river bathymetries.

Between the two formulas, the blue zoomed-in images highlight a location surrounding the river upstream to 1000 m downstream where the UF-formula river bathymetries are lower than the CMR-formula ones, resulting in greater flood extent here in the UF-formula datasets. This leads to that, in the UF-formula datasets, the flood extent variation appears not only in locations already totally flooded in the CMR-formula but also in new regions that are never flooded in the CMR-formula datasets. Consequently, there are more variations in flood extent in the UF-formula datasets compared to the CMR-formula datasets.

### 3.5 Comparison of variations in RMSEs

Figure 11 shows that, in both formulas, the variation in RMSEs of the slope dataset is the smallest, followed by those of flow, width, and combination datasets. In addition, the variations in RMSEs of the UF-formula datasets are larger than the

CMR-formula datasets. These trends correspond to the order of variations in river bathymetries as mentioned in Section 3.1 and 3.2.

The blue and red dashed lines represent the RMSEs for the CMR and UF formulas when using the LiDAR-derived and NIWA-estimated parameters without adding any errors. Each line stands in the middle of boxplots of each formula, demonstrating that these parameters still contain some errors deviating the results from the true predictions. Apart from that, we also noted the UF-formula RMSEs are slightly higher than the CMR-formula ones. To explain, the CMR is developed for coarse-grained rivers like the Waikanae River, leading to lower RMSEs than the UF formula. In contrast, the UF formula was not developed for any specific river types, which may contribute to its slightly higher RMSE. However, these small differences in RMSEs between the datasets using two formulas highlight a broad applicability of the UF formula on rivers without categorising their types.

## 4   Discussion

Our research went a step further than previous studies (Durand et al., 2008; Lee et al., 2018; Moramarco et al., 2019; Kechnit et al., 2024) to quantify the uncertainty in flood predictions due to the errors in the estimated river bathymetry. In this research, we applied the Monte Carlo method to generate a large number of simulations to capture the typical variability in the flood predictions and included spatial variability in our method. Moreover, we not only considered associated error distributions in parameters collectively, but we also performed a sensitivity analysis to assess the impact of each parameter. Hence, for situations where we lack the river bathymetric data and cannot collect or measure them for flood modelling, the formulas in this study can be used with the Monte Carlo assessment here that shows the sensitivity and understanding of the limitations and uncertainties involved. Furthermore, the analysis framework in this research can be applied to a wide range of formulas, such as those that also consider the sediment impacts, that are used to estimate river bathymetries to represent rivers in the flood modelling. Future research about this can help to answer which formula contributes the most to the uncertainty in flood model outputs.

In our research, we enhance the applicability of our findings by using the UF formula which is not constrained by specific observed data and applicable across a wide range of river types. Our results, based on the slight differences in the RMSEs between the datasets using the CMR and the UF equations, suggest the general applicability of the UF formula without the need of river categorisation. However, because we have only compared the UF formula with the CMR developed for coarse-grained rivers, comparisons with other formulas and approaches are still needed to confirm the applicability of the UF formula.

The results of our research can help the data collection process in which the parameters that have the greatest impact (specifically river flow and width) should be focused on measuring if resources are limited. Meanwhile, the parameter associated with the lowest influence (river slope) can be deprioritised. Nevertheless, due to the time-intensity and complexity, we have not explored the errors in the river Manning's n as well as $\alpha$ and $\beta$ coefficients. Furthermore, the Waikanae River bank-full flow is not strongly correlated with the variability of the bathymetry along the river as it stays nearly constant. This is based on the fact that the Waikanae River sections in our paper were not joined by major tributaries. Hence, future studies

should investigate the errors associated with these factors and perform a thorough sensitivity analysis to better support the data collection process.

In practice, different rivers will have different characteristics. Hence, it is necessary to generalise this study by considering a wide range of rivers for comparison and confirmation for the results found here. Accordingly, further research focusing on many rivers with diverse features is recommended.

Due to the lack of information about the sources of errors, the expected errors in our research were assumed to be unbiased and normally distributed with zero mean and a standard deviation of 10% of the best-estimated values. Hence, different realistic sources of errors should be considered to compare their impacts on the flood predictions. However, owing to the time intensity and complexity, this issue should be researched in another study.

Nguyen et al. (2025) analysed how the grid resolution influences on the flood predictions, which was not considered in our study. The change in grid size can cause a significant change in the river bathymetry and flood results. To capture the river structure with high accuracy, the grid resolution should provide several grid cells across the river. This ensures the river is well resolved for flood modelling. Accordingly, a further study about this is essential for better understanding.

Using the UF and CMR formulas with the best estimated parameters to obtain the river bathymetry can overcome the time-intensive and expensive nature of its acquisition. However, it contains the errors which can affect the flood predictions as our paper analysed above. Currently, without using Monte Carlo framework, a freeboard is often added to the flood level for addressing such uncertainty. It typically considers deviations in flood estimate, construction tolerances, and natural factors not accounted for in the calculations (Ministry for the Environment, 2024). However, this technique does not account for variations in flood extents, as demonstrated in Section 3.4 in this research, which can be influenced by the estimated river bathymetry. This suggests a future investigation to improve the effectiveness of this technique.

On the other hand, although applying the Monte Carlo framework to quantify this uncertainty is fully comprehensive, its requirement of a large amount of simulations can be seen as a computationally expensive problem. Due to this, the uncertainty quantification is not normally considered in the flood risk management. Hence, a more computational efficient method is essential. Machine learning approach, well-known for its more effective process to obtain the comparable results, is a good candidate which needs further investigation.

## 5  Conclusions

Our research focused on quantifying the uncertainty in flood predictions due to the errors in parameters used to estimate the river bathymetries. We applied LISFLOOD-FP flood model within a Monte Carlo method to generate multiple flood simulations for the January-2005 Waikanae River flood event for analysis. We performed a sensitivity analysis on three estimated parameters (river slope, flow, and width) and two formulas (the UF and CMR formulas) to assess their error impacts on the flood predictions individually and collectively through the estimated river bathymetries.

We found that, among three parameters, the uncertainty in flood model outputs, when the errors were added into the river width, is higher than when the errors were added into the river flow, followed by the river slope. The combination of all of them

was found to have the highest uncertainty. Between two formulas, the uncertainty in the flood predictions, especially in the flood depths and extents, when using the UF formula for estimating the river bathymetric data, is larger than using the CMR formula.

It is recommended that, instead of developing from scratch, the Monte Carlo framework used for the sensitivity analysis in this research should be applied to benchmark various formulas used to estimate the river bathymetries to represent rivers in flood modelling. Further study is necessary to confirm the broad applicability of the UF formula without river categorisation. Moreover, based on our results, the data collection process should focus on measuring the parameters (river width and flow) that have more significant impacts on the flood predictions if the resources are limited. Additionally, further investigations

should also include the river Manning's n, and $\alpha$ and $\beta$ coefficients to perform a thorough sensitivity analysis.

Apart from that, we suggested another study to be implemented on many rivers with different features. In addition, further research should consider how different realistic sources of errors affect the flood predictions. Also, the impacts of grid resolution on the estimated river bathymetry and on the flood predictions should be focused in future study. Currently, to cover such uncertainty, a freeboard is often used, but it fails to cover the variation in the flood extent, and thus a further study

is recommended to improve its effectiveness. Lastly, there is a need for simpler and faster methods than the Monte Carlo framework such as machine learning approaches to be included in flood risk management.

*Code and data availability.* The project employed GeoFabrics package and the flood model LISFLOOD-FP developed by Pearson et al. (2023) and Bates et al. (2010) respectively. All necessary data for simulation generation and results have been saved at Nguyen et al. (2024)

**Appendix A: Explanation about parameter exponents**

The exponents mentioned in Section 2.2. are exponents of each parameter after being processed from the original $\alpha$ and $\beta$. Specifically, for the UF formula, with $\alpha = 2/3$ and $\beta = 1/2$, it can be processed as below:

$$h = \left(\frac{nQ}{wS^{1/2}}\right)^{\frac{1}{1+2/3}} \Leftrightarrow h = \left(\frac{nQ}{wS^{0.5}}\right)^{0.6} \Leftrightarrow h = \frac{n^{0.6}Q^{0.6}}{w^{0.6}S^{0.3}} \tag{A1}$$

Hence, the exponents of the slope (S), bankfull flow (Q), and width (w) for the UF formula are 0.3, 0.6, and 0.6. For the CMR formula, with $\alpha = 0.745$ and $\beta = 0.305$, it can be changed as below:

$$h = \left(\frac{nQ}{wS^{0.305}}\right)^{\frac{1}{1+0.745}} \Leftrightarrow h = \left(\frac{nQ}{wS^{0.305}}\right)^{0.573} \Leftrightarrow h = \frac{n^{0.573}Q^{0.573}}{w^{0.573}S^{0.175}} \tag{A2}$$

Hence, the exponents of the slope (S), bankfull flow (Q), and width (w) for the CMR formula are 0.175, 0.573, and 0.573.

*Author contributions.* Based on the Contributor Role Taxonomy: Conceptualization - MN, MW, and EL; Funding acquisition - MW and EL, Formal analysis - MN, Data curation - MN, Methodology - MN, MW, EL, JB, and RP, Software - MN, RP, Supervision - MW, EL, and JB, Writing original draft - MN, Writing - review & editing - MW, EL, JB, and RP

*Competing interests.* The authors declare that they have no conflict of interest.

*Acknowledgements.* The project was sponsored by the Ministry for Business, Innovation and Employment as a PhD scholarship awarded to Martin Nguyen through the New Zealand flood hazard assessment programme, "Reducing flood inundation hazard and risk across Aotearoa – New Zealand". It is led by NIWA in collaboration with Geospatial Research Institute, School of Earth and Environment, University of Canterbury in Christchurch, New Zealand.

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
