# Peer review of "Quantifying uncertainty in flood predictions due to river bathymetry estimation"

_Hydrology and Earth System Sciences, 2024_

## Author Comment (AC1)

**Responses to the first reviewer about: "Quantifying uncertainty in flood predictions due to river bathymetry estimation"**

Martin Nguyen[1, 2, 3], Matthew D. Wilson[1, 3], Emily M. Lane[4], James Brasington[2, 3], and Rose A. Pearson[4]

[1]Geospatial Research Institute, University of Canterbury, Christchurch, New Zealand
[2]Waterways Centre, University of Canterbury, Christchurch, New Zealand
[3]School of Earth and Environment, University of Canterbury
[4]National Institute of Water and Atmospheric Research (NIWA), Christchurch, New Zealand

**Correspondence:** Martin Nguyen (tmn52@uclive.ac.nz)

Dear editorial support team and reviewer,

We thank you so much for the opportunity to reply to the reviewer about "Quantifying uncertainty in flood predictions due to river bathymetry estimation". Please see below, in green, for our responses.

***Summary of reviewer's comments:*** The reviewer was asking for further information and clarification about the relation between the river bathymetry and flood predictions, the flood model used in the paper, the question that the paper tries to answer, and the difference between two formulas used to estimate the river bathymetry.

***Summary of authors' responses:*** We thank you so much for helping us point out where we lack clarification and need improvement. In general, we summarised from the paper the relation between the river bathymetry and flood predictions through a simple chain in the response. We have added further information about LISFLOOD-FP - the flood model used in the paper. Also, we have clarified which question we are trying to fill in. Finally, we have shown the difference between two formulas with explanation.

**1 Question 1**

***Question:*** The relation between flood predictions and river bathymetry estimation is not clear for me. When talking about flood prediction, I would expect the simulation or prediction of streamflow, the peak flow volume and peak time. However, reading the paper, I only find results of bathymetry and flood extents. The authors didn't provide many details about the calculation of flood extents, but it seems that the flood extents can be determined by the river bathymetry directly. Consequently, in my understanding, it seems that "flood prediction" and "bathymetry estimation" in this study is one thing to some extent.

***Answer:*** River bathymetry refers to the river depth measurement. It plays a crucial role in flood modelling because it determines when and where water leaves the river channel and starts to flood overland. Based on this, the flood extent and flood depth, controlled by the topography outside of the river and the amount of the water that leaves the river, can be affected. However, measuring the river bathymetric data using some current methods like swath beam sonar and blue-green LiDAR is time-consuming, expensive, and sometimes unable if the water is deep or sediment-laden. Hence, various approaches have been proposed to estimate these river bathymetric data. This information was mentioned between lines 18-24. In that, more

information has been added between lines 22-23: *"Multi-beam sonar is effective but time-consuming, while blue-green LiDAR is faster but unable to obtain measurements in sediment-laden or deep water, and both of them are expensive.".*

Due to the inability to capture thoroughly the randomness of the river systems, these estimations contain uncertainties. If these estimated river bathymetric data are used to calculate the riverbed elevations to represent the river in topographic data like DEM and Manning's n (converted from roughness length) and use these data in a flood modelling, the flood model predictions will be affected. The flow of logic for this paper is as below:

Estimated river bathymetric data → riverbed elevations → topographic data (DEM and roughness length/Manning's n) generated by riverbed elevations and LiDAR data → inputs to a flood model (LISFLOOD-FP in this study) → affects flood model outputs

This information was written between lines 52-55 in the Introduction section, but it has been rewritten for clarification: *"Regardless of any approaches, due to the inability to capture the randomness of the river systems, errors in the measurements or estimations can introduce uncertainties that significantly deviate the simulated river bathymetries from the actual ones. Consequently, using these modelled river bathymetries to represent the river in flood modelling can affect the flood predictions. Currently, only limited studies concentrate on how these errors in bathymetry estimation can have impacts on the flood model outputs (Durand et al., 2008; Lee et al., 2018; Moramarco et al.,2019; Kechnit et al., 2024).".*

In our paper, we used the LISFLOOD-FP, a 2D hydrodynamic flood model to model the flood map/flood extent forced by a hydrograph (as seen in Figure 1b in the paper). Its inputs include topographical data (DEMs and Manning's n maps) and January-2005 flow/hydrograph and tidal data (kept fixed throughout all simulations). In the topographical data, as mentioned above, the river is represented by the riverbed elevations calculated from the estimated river bathymetric data and the surrounding land is represented by the LiDAR-derived topography.

Among the flood model outputs/flood predictions, the maximum water depths (MWDs) (rather than forecasting streamflow or peak timing) are selected to analyse how they are influenced by the uncertainties in the estimated river bathymetric data. This flood model output was chosen because its variation through simulations can be easily manipulated and visualised. For each dataset of 50 MWDs, we calculated the mean (mMWDs), standard deviation (sdMWDs), and coefficient of variation (covMWDs) of MWDs. We also calculated proportion of simulations in which a given pixel was flooded (pFs) to distinguish where was always flooded, never flooded, and sometimes flooded throughout these realisations. In this study, the mMWDs and sdMWDs did not add insight, so we did not consider them. This whole information was mentioned between lines 163-169. As to the flood extents, thank you so much for helping us to point this out, its information has been added at line 169: *"The flood extents were then calculated based on these pFs by multiplying the area of one pixel (10 m x 10 m) with number of pixels that were always and sometimes flooded.".*

**2 Question 2**

**Question:** Related to the first comment, I am also confused about the flood prediction model used in this study. The authors provide little descriptions on the LISFLOOD-FP model, missing some important issues. For example: what is the input and

output of the model? What's the relationship between this model and the two formulas for river bathymetry estimation? Is the estimated river bathymetry used in this model?

*Answer:* The information about LISFLOOD-FP model was shortly mentioned between lines 105-106: *"These DEM and Manning's n were then applied to the LISFLOOD-FP flood model (Bates et al., 2010; Neal et al., 2018), which was calibrated for this site in Nguyen et al. (2024b), to run the flood simulations."*. However, for clarification, more information have already been added from line 105: *"In this study, LISFLOOD-FP model, a 2D hydrodynamic model, is chosen to simulate the January-2005 flood event because it is well known for its computational efficiency and highly accurate flood model outputs (Nguyen et al., 2025). In this flood model, the formula to compute the water flow Qcell between cells at index $i$ over a time step $\Delta t$ is:*

$$Qcell_{i+1/2}^{t+\Delta t} = \frac{q_{i+1/2}^t - gh_{flow}^t \Delta t Scell_{i+1/2}^t}{\left[1 + \frac{g\Delta t n^2 |q_{i+1/2}^t|}{(h_{flow}^t)^{7/3}}\right]} \Delta x \qquad (1)$$

where $q^t$ represents the flux at time $t$, $\Delta x$ denotes the cell width, $Scell$ and $h_{flow}$ are the water surface slope and flow depth between cells (Bates et al., 2010). The flow formula here is displayed for the x direction, the y direction can be obtained analogously. The cell water depth h is updated based on the discharge through the four boundaries of that cell as below, where $i$ and $j$ denote the cell coordinates (Shustikova et al., 2019):

$$\frac{\Delta h^{i,j}}{\Delta t} = \frac{Qcell_x^{i-1,j} - Qcell_x^{i,j} + Qcell_y^{i,j-1} - Qcell_y^{i,j}}{\Delta x^2}. \qquad (2)$$

This flood model was calibrated for the Waikanae River site in Nguyen et al. (2025)."*.

The main inputs for the LISFLOOD-FP model to simulate the January-2005 flood event are the river flow data, tidal data, topographical data that include the estimated river bathymetry - DEM and Manning's n converted from roughness length using the equation 1 mentioned in the paper. The main outputs of the model are water surface elevation and water depth across the time series and their maximum values. Among them, the study chose the MWDs to analyse as mentioned in question 1. The information about these inputs and outputs was mentioned in the paper but has been rewritten between lines 163-164 for clarification: *"These DEMs and Manning's n maps that include the simulated river bathymetric data, along with January-2005 flow and tidal data, were then used in the LISFLOOD-FP flood model to produce 50 maximum water depths (MWDs) for further statistical analysis."*. In this, Nguyen et al. (2024b) is now published as Nguyen et al. (2025).

To expand the flow of logic mentioned in the question 1, here we added more details on how the uncertainties in the parameters used to estimate the river bathymetric data propagate through the LISFLOOD-FP flood model to the outputs as below:

Estimated parameters that include uncertainties (river slope, width, and flow) → two chosen formulas to estimate river bathymetric data → riverbed elevations → topographic data (DEM and Manning's n) → inputs to the LISFLOOD-FP flood model → affects maximum water depth.

According to the above chain, we assumed the uncertainties in the estimated river bathymetric data arising from the estimated parameters (river slope, flow, and width) used in two chosen formulas - Conceptual Multivariate Regression (CMR)

and Uniform Flow (UF) (mentioned between lines 119-120). These river bathymetric data were then subtracted from the LiDAR-estimated water surface elevation to obtain the riverbed elevations (mentioned lines 158-160). These riverbed elevations, along with the topographic LiDAR data collected from the OpenTopography, were then sampled and interpolated onto a square grid to obtain topographic data like DEM and roughness length/Manning's n (mentioned lines 96-101, 161-163). The roughness length/Manning's n was then used in the LISFLOOD-FP flood model to produce flood model outputs. As mentioned before, we selected the MWDs among these outputs for uncertainty analysis for easy data manipulation.

**3 Question 3**

As this question includes a lot of small queries, we divided it into two sub-questions to answer as below:

*Question:* The uncertainties in flood predictions come from several sources, including the accuracy of three parameters (S, Q and w), the accuracy of two formulas for bathymetry estimation, and the influence of bathymetry uncertainties on flood prediction. Unfortunately, the analysis conducted in this study is more likely a sensitivity analysis, without addressing these uncertainty sources clearly. The authors analyzed the uncertainty brought by a standard deviation of 10%, but the question should be what is the actual uncertainty in S, Q and w estimation themselves

*Answer:* Regardless of any approaches to estimate the river bathymetric data, due the inability to capture the randomness of the real-world river systems, these estimations still contain errors. If they are used to represent the rivers in the topographic data like DEM which is an input for a flood modelling, the flood predictions will be affected. Based on this, our focus is to quantify the uncertainty in the flood model outputs due to such errors in the estimated river bathymetric data. This will help us to answer the question: "How significant is the uncertainty in the flood predictions arising from the errors in the estimated river bathymetric data?". There have been some papers that tried to answer this question, but there are certain gaps in their methodologies. This idea was written between lines 52-55 and lines 75-78. In that, lines 52-55 have been rewritten for clarification: *"Regardless of any approaches, due to the inability to capture the randomness of the river systems, errors in the measurements or estimations can introduce uncertainties that significantly deviate the simulated river bathymetries from the actual ones. Consequently, using these modelled river bathymetries to represent the river in flood modelling can affect the flood predictions. Currently, only limited studies concentrate on how these errors in bathymetry estimation can have impacts on the flood model outputs (Durand et al., 2008; Lee et al., 2018; Moramarco et al.,2019; Kechnit et al., 2024)."*.

In this paper, because we have no information about what the uncertainty is, we look into the sensitivity. Specifically, we developed a sensitivity analysis using Monte Carlo framework which can be used for different formulas and parameters. Within this framework, we chose two formulas that have been validated and used to estimate the river bathymetry at the Waikanae River - the UF and CMR - by Pearson et al. (2021). Due to the time intensity and complexity, we selected three parameters - river slope, flow, and width - and examined how their errors propagate through the flood modelling and affect the outputs. This idea was written between lines 79-82, and have been rewritten for clarification: *"In this paper, we quantified the uncertainty in flood predictions due to errors in estimated parameters used in two formulas described in Rupp and Smart (2007) and Neal et al.*

*(2021). These two methods were validated by Pearson et al. (2021). Within the Monte Carlo framework, we generated multiple realisations of river bathymetry, then used this to perform a sensitivity analysis to evaluate the impacts of each parameter on flood predictions, individually, and collectively.".*

The errors of each parameter were drawn from a normal distribution with zero mean and a standard deviation set to 10% of the best estimate of that parameter. We chose that 10% for all parameters because: (i) several observed cross-sectional riverbed elevations fall within the area of simulated riverbed elevations - calculated from the simulated river bathymetric data as seen in Figure 1 of this answer sheet; (ii) with the same amount of errors, we can then compare the impacts of those errors, between datasets, on the flood predictions. Generally, this helps us to see, with this small amount of errors, how they propagate through the flood modelling and affect the flood model outputs. Higher levels of errors such as 20%, 30%, etc. should also be considered, but due to the time intensity and complexity, another research would be a better fit. This information and the Figure 1 in this answer sheet will be added into the paper.

The main results have shown that the errors in the estimated river bathymetric affect at some level to the flood model outputs. In particular, between two formulas, the errors in the parameters using the UF formula are associated with greater uncertainty in flood predictions than the CMR formula. However, when we validated them with the observed flood data, their RMSEs are not much different. This suggested the applicability of the UF formula to estimate the river without the need of river categorisation. Nevertheless, further research is needed to compare the UF formula with other approaches. This has been shown in Section 3.3, 3.4, and 3.5. It has also been mentioned between lines 335-337, but it has been rewritten for clarification: *"... However, because we have only compared the UF formula with the CMR developed for coarse-grained rivers, other comparisons with other formulas still need further research to confirm the applicability of the UF formula.".*

Between parameters, the uncertainty in flood model outputs associated with the river slope parameter is the smallest, followed by the river flow, and width. This information can support the data collection process when the resources are limited. Specifically, we can focus on measuring the parameters that have the greatest impacts (river flow and width) and deprioritize the one associated with the lowest influences (river slope). However, as mentioned above, due to the time intensity and complexity, we have not explored the errors in the river friction as well as $\alpha$ and $\beta$ coefficients. This has been shown in Section 3.3, 3.4, and 3.5. This point was also mentioned between lines 338-341 and has been rewritten for clarification: *"The results of our research can help the data collection process in which the parameters that have the greatest impact (specifically river flow and width) should be focused on measuring if resources are limited. Meanwhile, the parameter associated with the lowest influence (river slope) can be deprioritised. Nevertheless, due to the time-intensity and complexity, we have not explored the errors in the river friction as well as $\alpha$ and $\beta$ coefficients. Furthermore, the Waikanae River bank-full flow is not strongly correlated with the variability of the bathymetry along the river as it nearly stays constant. Hence, future studies should investigate the errors associated with these factors and perform a thorough sensitivity analysis to support the data collection process better.".*

Generally, our focus is to find out how the errors in estimated river bathymetric data can affect the flood model outputs. Since we have no information about what the uncertainty is, we look into the sensitivity. Furthermore, there are many sources of errors for such parameters and they would be different for different formulas. Hence, we developed this sensitivity analysis

[Figure]

**Figure 1.** Observed cross-sections, best estimates, and simulations of riverbed elevations at the Waikanae River: the Uniform Flow formula - (a) slope, (c) bank-full flow, (e) width, and (g) combined; the Conceptual Multivariate Regression formula - (b) slope, (d) bank-full flow, (f) width, and (h) combined.

using the Monte Carlo framework that can be applicable to various formulas and parameters to assess which parameters that have errors can affect significantly on the flood model outputs. This will support the data collection process to focus on these parameters if the resources are limited and also suggest future investigations to research on errors in these parameters.

*Question:* Besides, the study didn't use any measurement data to validate the estimated bathymetry, so the analysis actually only shows the range of estimated bathymetry caused by a 10% variation in S/Q/w, which, in my opinion, is a rather direct procedure from the viewpoint of mathematic, since the formulas for bathymetry (Eq.2) is a very simple equation.

The estimated river bathymetry for the Waikanae River was already validated by Pearson et al. (2021). As mentioned above, the errors for each parameter were selected from a normal distribution with zero mean and a standard deviation set to 10% of the best estimate of that parameter. Based on this and to also consider the spatial variability along the river, the Gaussian variograms were applied to generate unconditional simulations for each parameter. This idea has been written and explained in Section 2.4 between lines 138-155. As the reviewer said, this sensitivity of the 10% change on the equations is simple to compute, but understanding how that then affects the flood extent is not straightforward and that is what this manuscript investigates. In other words, this analysis provided information about how the errors in the estimated river bathymetric data can propagate and affect the flood model outputs.

**4   Question 4**

*Question:* Some questions about UF and CMR formulas. 1) According to section 2.2 the only difference in these two formulas is the different value of $\alpha$ and $\beta$, am I right? 2) Table 1: In my understanding, Manning's n should be a parameter reflecting the characteristics of riverbed. Why is it different in different formulas?

*Answer:* The two formulas have different values of $\alpha$, $\beta$, and river Manning's n. The Conceptual Multivariate Regression formula has a constant river Manning's n because it is developed specifically for coarse-grained rivers. To highlight this idea, the information between lines 117-118 can be rewritten: *"For the $\alpha$ and $\beta$ coefficients, the UF formula used two constant values of 2/3 and 1/2, while the CMR formula, designed for coarse-grained rivers, applied 0.745 and 0.305 respectively with another constant value of 0.162 for Manning's n."*.

---

## Author Comment (AC2)

**Responses to the second reviewer about: "Quantifying uncertainty** in flood predictions due to river bathymetry estimation"**

Martin Nguyen1, 2, 3, Matthew D. Wilson1, 3, Emily M. Lane4, James Brasington2, 3, and Rose A. Pearson4

1Geospatial Research Institute, University of Canterbury, Christchurch, New Zealand
2Waterways Centre, University of Canterbury, Christchurch, New Zealand
3School of Earth and Environment, University of Canterbury
4National Institute of Water and Atmospheric Research (NIWA), Christchurch, New Zealand

Correspondence: Martin Nguyen (tmn52@uclive.ac.nz)

Dear editorial support team and reviewer,

We thank you so much for the opportunity to reply to the reviewer about "Quantifying uncertainty in flood predictions due to river bathymetry estimation". Please see below, in green, for our responses.

Summary of reviewer's comments: The reviewer was asking for further information and clarification about the flood model
used in the paper, adequate assessment for the uncertainty in flood model outputs, summary of previous publication - Nguyen et al. (2024b), context of this uncertainty analysis, and more case studies for robustness.

*Summary of authors' responses:* We thank you so much for helping us point out where we lack clarification and need improvement. In general, we have added further information about LISFLOOD-FP - the flood model used in the paper. In this answer sheet, we have also indicated how we analysed the flood model outputs in the paper and summarised the previous

10 publication - Nguyen et al. (2024b) which is now published as Nguyen et al. (2025). We have finally explained the robustness of our paper and suggest that another study would be better to investigate a wide range of rivers.

**1 Question 1**

*Question:* Generally, the authors need to significantly improve the methods section to convey the methods used in this study. In particular, they did not provide enough explanation regarding the LISFLOOD modeling and the input data utilized. It

15 is important to summarize the processes implemented by the model to understand the relationships presented in the results section. For example, processes such as, backwater effect, sediment processes, human regulations, etc.

Answer: The information about LISFLOOD-FP model was shortly mentioned between lines 105-106: "These DEM and Manning's n were then applied to the LISFLOOD-FP flood model (Bates et al., 2010; Neal et al., 2018), which was calibrated for this site in Nguyen et al. (2024b), to run the flood simulations.". However, for clarification, more information have already

20 been added from line 105: "In this study, LISFLOOD-FP model, a 2D hydrodynamic model, is chosen to simulate the January-2005 flood event because it is well known for its computational efficiency and highly accurate flood model outputs (Nguyen et al., 2025). In this flood model, the formula to compute the water flow, Qcell, between cells at index i over a time step  $\Delta t$  is:

$$Qcell_{i+1/2}^{t+\Delta t} = \frac{q_{i+1/2}^t - gh_{flow}^t \Delta tScell_{i+1/2}^t}{\left[1 + \frac{g\Delta tn^2 |q_{i+1/2}^t|}{(h_{flow}^t)^{7/3}}\right]} \Delta x$$
(1)

where qt represents the flux at time t, Δx denotes the cell width, Scell and hflow are the water surface slope and flow
25 depth between cells (Bates et al., 2010). The flow formula here is displayed for the x direction, the y direction can be obtained analogously. The cell water depth h is updated based on the discharge through the four boundaries of that cell as below, where i and j denote the cell coordinates (Shustikova et al., 2019):

$$\frac{\Delta h^{i,j}}{\Delta t} = \frac{Qcell_x^{i-1,j} - Qcell_x^{i,j} + Qcell_y^{i,j-1} - Qcell_y^{i,j}}{\Delta x^2}.$$
(2)

This flood model was calibrated for the Waikanae River site in Nguyen et al. (2025).".

- 30 The main inputs for the LISFLOOD-FP model to simulate the January-2005 flood event are the river flow data, tidal data, DEM, and Manning's n converted from roughness length. The main outputs of the model are the water surface elevation and water depth across the time series and their maximum values. Among them, the study chose the maximum water depths (MWDs) to analyse. The information about these inputs and outputs was mentioned between lines 163-164, but has been rewritten: "*These DEMs and Manning's n maps that include the simulated river bathymetric data, along with January-2005*
- **35** flow and tidal data, were then used in the LISFLOOD-FP flood model to produce 50 maximum water depths (MWDs) for further statistical analysis.".

According to Bates et al. (2010), the LISFLOOD-FP flood model does not assume uniform flow and includes the surface slope, *Scell*, which allows the model to simulate the situations like backwater effects - where the water flows uphill or slows down due to downstream resistance like tides. However, the flood model does not include sediment processes. Also, for human

40 regulations, it partially supports by representing the levees/embankments, for example, through the DEM or manually inserted structures. In this study, the LISFLOOD-FP flood model and our research focuses on the pixel-level hydrodynamics and spatial water depth. Hence, the processes including backwater effect, sediment processes, and human regulations are not our main focus.

**2 Question 2**

50

45 We divided it into two sub-questions to answer as below:

*Question:* I do not believe the authors adequately assess the uncertainty of floods, as they primarily evaluate the uncertainty of DEMs, bathymetry estimations, and roughness coefficients.

*Answer:* The reviewer is correct, we do not assess all the uncertainties in floods as this would be an incredibly big task. As we clearly stated in the Introduction section, or especially between lines 75-84, we are focussing on this one particular aspect of uncertainty which can then be added to studies of other sources of uncertainty to understand the bigger picture. We now

briefly explain how we analyse this uncertainty in flood predictions arising from the estimated river bathymetry.

The variability of simulations of the topographic data like DEM and roughness length/Manning's n gather around the river due to the simulated river bathymetric data. Hence, we only focus on the variability of the simulated river bathymetric data as seen in Figure 5 in the paper. This information has been added between lines 161-164: *"Similar to Durand et al. (2008);*

- 55 Moramarco et al. (2019); Kechnit et al. (2024), and especially Nguyen et al. (2024b), our research also applied a Monte Carlo framework to generate 50 DEMs and 50 Manning's n maps with 50 different bathymetries using the method mentioned in Section 2.1 for each dataset. Since the variability of simulations of these topographic data gathers around the river, we only focus on the variation in the simulated river bathymetric data. The simulated topographic data including DEMs and Manning's n maps were then used for modelling the January-2005 flood event to produce 50 maximum water depths (MWDs) for further
- 60 statistical analysis.".

After that, we analysed how the variability in the river bathymetric data can affect the flood model outputs - maximum water depths (MWDs). Specifically, for each of eight datasets of 50 MWDs, we computed their mean (mMWDs), standard deviation (sdMWDs), and coefficient of variation (covMWDs). Here, because the mMWDs and sdMWDs did not provide further insights, they are not considered in the paper. We also calculated proportion of simulations in which a given pixel was

65 flooded (pFs) to distinguish where was always flooded, never flooded, and sometimes flooded throughout these realisations. For the flood extent, its calculation information has been added at line 169: *"The flood extents were then calculated based on these pFs by multiplying the area of one pixel (10 m x 10 m) with number of pixels that were always and sometimes flooded."*. Apart from this, we also validated each flood simulation with the observed flood data using the RMSE metric.

The results of covMWDs, flood extents, and RMSEs were shown in Figures 6-7, Figures 8-9, and Figure 10. This corresponds to the Sections 3.3, 3.4, and 3.5. In particular, we used boxplots (Figures 6, 8, 10) to compare the variations of eight datasets

70 to the Sections 3.3, 3.4, and 3.5. In particular, we used boxplots (Figures 6, 8, 10) to compare the variations of eight datasets and maps (Figures 7 and 9) to visualise and explain. Here, our explanation linked with what we found in the variations of simulated river bathymetric data.

Generally, based on those Figures, we found that the variations in the MWDs based on covMWDs and flood extents correspond to the variations in the estimated river bathymetries. In particular, with the same amount of uncertainty added to

- 75 each of eight datasets, the variation in the slope parameter corresponds to the smallest variation in the MWDs, followed by the flow and the width. Between two formulas, the errors in the parameters of the UF formula are associated with greater uncertainty in the MWDs than those of the CMR formula. We provided the explanation as below for each Section:
  - For covMWDs, Section 3.3, lines 290-295: "To explain, between parameter datasets, the small variability in the river bathymetry corresponding with the variation in the river slope does not significantly affect the water spreading into the
- 80 floodplain, unlike the variations in the river bank-full flow and width. The impacts of all these variations become more apparent in floodplains farther from the river, especially at flood boundaries in midstream, where the water has less direct connection with the river. Between two formulas, because the variations in the UF-formula river bathymetries are higher than the CMR-formula ones as seen in Fig. 5, the variations in the flood depths of the UF-formula datasets are also higher than the CMR-formula datasets.".

- For flood extents, Section 3.4, lines 307-312: "Between the two formulas, the blue zoomed-in images highlight a location surrounding the river upstream to 1000 m downstream where the UF-formula river bathymetries are lower than the CMR-formula ones, resulting in greater flood extent here in the UF-formula datasets. This leads to that, in the UF-formula datasets, the flood extent variation appears not only in locations already totally flooded in the CMR-formula but also in new regions that are never flooded in the CMR-formula datasets. Consequently, there are more variations in flood extent in the UF-formula datasets compared to the CMR-formula datasets.".
  - For RMSEs, Section 3.5, lines 312-325: "To explain, the CMR is developed for coarse-grained rivers like the Waikanae River, leading to lower RMSEs than the UF formula. In contrast, the UF formula was not developed for any specific river types, which may contribute to its slightly higher RMSE. However, these small differences in RMSEs between the two formulas highlight a broad applicability of the UF formula on rivers without categorizing their types.".
- 95 *Question:* In addition, the authors did not indicate whether their sources of uncertainty are valid by referring to the ranges of DEM values, any reported roughness, etc.

*Answer:* For each parameter, we selected the errors from a normal distribution with zero mean and standard deviation set to 10% of the best estimates of that parameter. This 10% was chosen because: (i) several observed cross-sectional riverbed elevations fall within the area of simulated riverbed elevations - calculated from the simulated river bathymetric data as seen in

100 Figure 1 in this answer sheet; (ii) with the same amount of errors, we can then compare the influences of those errors, between datasets, on the flood model outputs. This helps us to see, with this small amount of errors, how they propagate through the flood modelling and affect the flood predictions. Higher levels of errors such as 20%, 30%, etc. should also be considered, but due to the time intensity and complexity, another research would be a better fit. This information and the Figure 1 in this answer sheet will be added into the paper.

**105 3 Question 3**

*Question:* In addition, they refer to a previous publication, Nguyen et al. (2024b), to get the key details for the methods used. For a smooth reading experience, the authors should summarize that key information in this manuscript as well.

Answer: The summary of the similar key information in methodology to Nguyen et al. (2025) has been added at line 91 before the section 2.1.: "Our data and methodology were based on Nguyen et al. (2025) where the uncertainty in flood
predictions due to arbitrary conventions in grid alignment was quantified. To explain, their research is also about how the uncertainty in the process of generating the topographic data like DEM and roughness length can propagate through the flood

modelling to the outputs. Hence, their data and methodology can be applied in our research..

Accordingly, we simulated the same flood event using the LISFLOOD-FP flood model and applied a similar method to generate topographic data. Moreover, a Monte Carlo framework was also designed in our research to observe how the

115 uncertainty in estimated river bathymetries propagates through the flood modelling to the flood model outputs. To assess the

---

## Author Response (AR1)

**Responses to reviewers about: "Quantifying uncertainty in flood predictions due to river bathymetry estimation"**

Martin Nguyen[1, 2, 3], Matthew D. Wilson[1, 3], Emily M. Lane[4], James Brasington[2, 3], and Rose A. Pearson[4]

[1]Geospatial Research Institute, University of Canterbury, Christchurch, New Zealand
[2]Waterways Centre, University of Canterbury, Christchurch, New Zealand
[3]School of Earth and Environment, University of Canterbury
[4]National Institute of Water and Atmospheric Research (NIWA), Christchurch, New Zealand

**Correspondence:** Martin Nguyen (tmn52@uclive.ac.nz)

Dear Prof. Lixin Wang and reviewers,

We thank you so much for the opportunity to reply to the reviewers about "Quantifying uncertainty in flood predictions due to river bathymetry estimation". We appreciate the time and effort that you and the reviewers spent on providing valuable comments to our paper. Please see below, in green, for our responses to the reviewers' questions. The sections and lines

5   mentioned here are based on the track-changed manuscript version. Since some questions are about the same issue, we might provide similar answers. Apart from that, we added an extra section to list all of parts we edited to make the content more accurate, consistent, and concise. Additionally, please note that Nguyen et al. (2024b) is now published as Nguyen et al. (2025).

**1   Editor report**

10   Both reviewers think the topic is important. At the same time, both reviewers suggested that the lack of methodology details prevented a thorough evaluation of the novelty of this work, along with other major and minor concerns. I concur with the reviewers' assessment and would like to invite the authors for a thorough revision. Please keep the reviewers' comments in mind when carrying out the revision. The revised manuscript will be further reviewed.

We thank the editor again for this opportunity to revise our manuscript according to suggestions from the reviewers.

15   Generally, in the revised manuscript, we adjusted the Introduction Section to highlight the main focus of our paper and expanded the methodology section by mainly adding a summary of the previous paper - Nguyen et al. (2025), further information about Waikanae River, LISFLOOD-FP flood model, and our assumption of errors of 10%. In this response, we explain in detail the relation between river bathymetry and flood model outputs, how these flood model outputs were analysed, and the robustness of our paper. Also, the corresponding contents in the revised manuscript were modified for consistency.

**2 Reviewer 1**

***Summary of reviewer's comments:*** The reviewer was asking for further information and clarification about the relation between the river bathymetry and flood predictions, the flood model used in the paper, the research question that the paper tries to answer, and the difference between two formulas used to estimate the river bathymetry.

*Summary of authors' responses:* We thank you so much for helping us point out where we lack clarification and need improvement. In general, we summarised from the paper the relation between the river bathymetry and flood predictions through a simple chain in the response. We have added further information about LISFLOOD-FP - the flood model used in the paper. Also, we have clarified which research question we are trying to answer. Finally, we have shown the difference between the two formulas with explanation.

**2.1 Question 1**

***Question:*** The relation between flood predictions and river bathymetry estimation is not clear for me. When talking about flood prediction, I would expect the simulation or prediction of streamflow, the peak flow volume and peak time. However, reading the paper, I only find results of bathymetry and flood extents. The authors didn't provide many details about the calculation of flood extents, but it seems that the flood extents can be determined by the river bathymetry directly. Consequently, in my understanding, it seems that "flood prediction" and "bathymetry estimation" in this study is one thing to some extent.

*Answer:* For clarification, in this paper, flood predictions and flood modelling refers to inundation modelling rather than flood flows. River bathymetry refers to the river depth measurement. It plays a crucial role in flood modelling because it determines when and where water leaves the river channel and starts to flood overland. Based on this, the flood extent and flood depth, controlled by the topography outside of the river and the amount of the water that leaves the river, can be affected. However, measuring the river bathymetric data using some current methods like swath beam sonar and blue-green LiDAR is either time-consuming or expensive, and sometimes unfeasible if the water is deep or sediment-laden. Hence, various approaches have been proposed to estimate these river bathymetric data. This information was mentioned between lines 20-28. In that, more information has been added between lines 24-26: *"Multi-beam sonar is effective but time-consuming, while blue-green LiDAR is faster but does not work in sediment-laden or deep water, and both of them are expensive (Bailly et al., 2010; Flener et al., 2012; Bures et al., 2019)."*.

If these estimated river bathymetric data are used to calculate the riverbed elevations and represent the river in topographic data and then used in flood modelling, the model predictions will be affected. The flow of logic for this paper is as below:

Estimated river bathymetric data → riverbed elevations → topographic data (DEM and Manning's n derived from roughness length) generated by riverbed elevations and LiDAR data → inputs to a flood inundation model (LISFLOOD-FP in this study) → affects flood model outputs (extents, depths, etc.).

This information was written between lines 55-63 in the Introduction Section, but has been rewritten for clarification: *"Regardless of any approaches to estimate the river bathymetric data, due to the inability to capture the randomness of the real-world river systems, these estimations still contain errors. These errors can cause the simulated river bathymetries to*

*deviate significantly from the actual ones. Consequently, using these modelled river bathymetries to represent the rivers in food inundation modelling can affect the flood predictions. Currently, several studies have investigated the errors in the estimated river bathymetry (Durand et al., 2008; Lee et al., 2018; Moramarco et al.,2019; Kechnit et al., 2024), but they have not considered how these estimations with errors affect the flood model outputs. ".*

In our paper, we used the LISFLOOD-FP, a 2D hydrodynamic flood model, to model the flood map/flood extent forced by a hydrograph (as seen in Figure 1b in the paper). Its inputs include topographical data (DEMs and Manning's n maps) and January-2005 flow/hydrograph and tidal data (kept fixed throughout all simulations). In the topographical data, as mentioned above, the river is represented by the riverbed elevations calculated from the estimated river bathymetric data, and the surrounding land is represented by the LiDAR-derived topography.

Among the flood model outputs/flood predictions, the maximum water depths (MWDs) and maximum water surface elevations (MWSEs) (rather than forecasting streamflow or peak timing) were selected to analyse how they were influenced by the uncertainties in the estimated river bathymetric data. These flood model outputs were chosen because their variation through simulations can be easily manipulated and visualised. For each dataset of 50 MWDs, we calculated the mean (mMWDs), standard deviation (sdMWDs), and coefficient of variation (covMWDs) of MWDs. We also calculated proportion of simulations in which a given pixel was flooded (pFs) to distinguish where was always flooded, never flooded, and sometimes flooded throughout these realisations. In this study, the mMWDs and sdMWDs did not add insight, so we did not consider them. We also validated the flood simulations - MWSEs against observed flood levels using RMSE metric for the scenario modelled.

The whole above information was mentioned between lines 217-245. As to the flood extents, thank you so much for helping us to point this out, its information has been added between lines 234-237: *"Additionally, we computed expected flooded area or expected flood extent, a metric often employed by decision-makers, for each simulation for comparison. The expected flood extents were calculated based on these pFs by multiplying the area of one pixel (10 m x 10 m) with number of pixels that were always and sometimes flooded.".*

**2.2 Question 2**

*Question:* Related to the first comment, I am also confused about the flood prediction model used in this study. The authors provide little descriptions on the LISFLOOD-FP model, missing some important issues. For example: what is the input and output of the model? What's the relationship between this model and the two formulas for river bathymetry estimation? Is the estimated river bathymetry used in this model?

*Answer:* More information about LISFLOOD-FP model was added between lines 143-155:

*"In this study, LISFLOOD-FP (Bates et al., 2010; Neal et al., 2018), a 2D hydrodynamic model, was used to simulate the January-2005 flood event (which was calibrated for this site in Nguyen et al. (2025)) because it is well known for its computational efficiency and highly accurate flood model outputs (Nguyen et al., 2025). The DEM and Manning's n values, along with the flow information and tidal data mentioned above were used as input into this model.*

*In LISFLOOD-FP, the formula to compute the water flow Qcell at the interface index $i+1/2$, between cells index $i$ and index $i+1$, over a time step $\Delta t$ is:*

$$Qcell_{i+1/2}^{t+\Delta t} = \frac{q_{i+1/2}^t - gh_{flow}^t \Delta t Scell_{i+1/2}^t}{[1 + \frac{g \Delta t n^2 |q_{i+1/2}^t|}{(h_{flow}^t)^{7/3}}]} \Delta x \tag{1}$$

*where $q^t$ represents the flux at time t, $\Delta x$ denotes the cell width, Scell and $h_{flow}$ are the water surface slope and flow depth between cells (Bates et al., 2010). The flow formula here is displayed for the x direction, the y direction can be obtained analogously. The cell water depth $h_{flow}$ is updated based on the discharge through the four boundaries of that cell as below, where i and j denote the cell coordinates (Shustikova et al., 2019):*

$$\frac{\Delta h_{flow}^{i,j}}{\Delta t} = \frac{Qcell_x^{i-1,j} - Qcell_x^{i,j} + Qcell_y^{i,j-1} - Qcell_y^{i,j}}{\Delta x^2}." \tag{2}$$

As mentioned in question 1: The main inputs for the LISFLOOD-FP model to simulate the January-2005 flood event are the river flow data, tidal data, and topographical data that include the estimated river bathymetry - DEM and Manning's n converted from roughness length using the equation 1 mentioned in the Section 2.1. of the paper. The main outputs of the model are time series of water surface elevations and water depths and their maximum values. The study chose the MWDs and MWSEs to analyse. The information about these inputs and outputs is described in the paper between lines 217-245. In that, lines 217-226 were rewritten for clarification as below:

*"Similar to Durand et al. (2008); Moramarco et al. (2019); Kechnit et al. (2024), and especially Nguyen et al. (2025), our research also applied a Monte Carlo framework to generate 50 DEMs and 50 Manning's n maps from those 50 simulated rivebed elevations and LiDAR data from OpenTopography (2013) using the method described in Section 2.1 for each dataset. These 50 DEMs and 50 Manning's n maps are the same except for the river locations due to the use of 50 different simulated riverbed elevations. Hence, we only focus on analysing the variation in the simulated river bathymetric data used to generate these riverbed elevations instead of those simulated topographic data (see Section 3.2.). The DEMs and Manning's n maps that include the simulated river bathymetric data, along with the January-2005 flow and tidal data mentioned in Section 2.1, were then used in the LISFLOOD-FP flood model to produce 50 maximum water depths (MWDs) and 50 maximum water surface elevations (MWSEs) for further statistical analysis."*

To expand the flow of logic mentioned in question 1, we have added more details on how the uncertainties in the parameters used to estimate the river bathymetric data propagate through the LISFLOOD-FP flood model to the outputs as below:

Estimated parameters that include uncertainties (river slope, width, and flow) → two chosen formulas to estimate river bathymetric data → riverbed elevations → topographic data (DEM and Manning's n derived from roughness length) generated by riverbed elevations and LiDAR data → inputs to the LISFLOOD-FP flood model → affects flood extent and maximum water depth.

According to the above chain, we assumed uncertainties in the estimated river bathymetric data, in our case arising from the estimated parameters (river slope, flow, and width) and used these in two chosen formulas - Conceptual Multivariate

Regression (CMR) and Uniform Flow (UF) (mentioned at lines 168-169). The calculated river depths were then subtracted from the LiDAR-estimated water surface elevation to obtain the estimated riverbed elevations (mentioned at lines 135-138, 213-215). These riverbed elevations, along with the topographic LiDAR data collected from the OpenTopography, were then sampled and interpolated onto a square grid to obtain topographic data i.e., DEM and Manning's n (derived from roughness length) (mentioned at lines 146-147 and 223-226). These DEM and Manning's n were then used in the LISFLOOD-FP flood model to produce flood model outputs (mentioned at lines 137-139 and 212-216). As mentioned in question 1, we selected the MWDs and MWSEs among these outputs for uncertainty analysis.

**2.3 Question 3**

We divided this question into two sub-questions to answer as below:

*Question:* The uncertainties in flood predictions come from several sources, including the accuracy of three parameters (S, Q and w), the accuracy of two formulas for bathymetry estimation, and the influence of bathymetry uncertainties on flood prediction. Unfortunately, the analysis conducted in this study is more likely a sensitivity analysis, without addressing these uncertainty sources clearly. The authors analyzed the uncertainty brought by a standard deviation of 10%, but the question should be what is the actual uncertainty in S, Q and w estimation themselves.

*Answer:* Regardless of any approaches to estimate the river bathymetric data, due to the inability to capture the randomness of the real-world river systems, these estimations still contain errors. If they are used to represent the rivers in the topographic data like DEM which is an input for a flood modelling, the flood predictions will be affected. Based on this, our focus is to quantify the uncertainty in the flood model outputs due to such errors in the estimated river bathymetric data. This will help us to answer the research question: "How do the errors in the estimated river bathymetric data affect the flood model outputs?". Previous studies investigated the errors in river bathymetry estimations, but they did not evaluate how such errors could affect the flood model outputs. These whole ideas were mentioned in the Introduction Section lines 55-95 and rewritten for clarification as below:

– Lines 55-63: *"Regardless of any approaches to estimate the river bathymetric data, due to the inability to capture the randomness of the real-world river systems, these estimations still contain errors. These errors can cause the simulated river bathymetries to deviate significantly from the actual ones. Consequently, using these modelled river bathymetries to represent the rivers in food inundation modelling can affect the flood predictions. Currently, several studies have investigated the errors in the estimated river bathymetry (Durand et al., 2008; Lee et al., 2018; Moramarco et al.,2019; Kechnit et al., 2024), but they have not considered how these estimations with errors affect the flood model outputs."*

– Lines 64-73: *"For instance, Durand et al. (2018) developed an ensemble-based data assimilation approach for estimating river bathymetry from water surface elevation measurements and the LISFLOOD-FP hydrodynamic model. Using a Monte Carlo-based framework, they also performed a sensitivity analysis to assess how various error sources affected the estimated results. Their study found that errors in some input factors for their approach, such as river roughness and flow conditions, have greater influence than the water surface elevation measurement errors. However, this research did*

150      *not evaluate how the errors in these river bathymetric estimations can affect flood model outputs with consideration of spatial variability of input factors in the analysis."*

-    Lines 79-83: *"Nevertheless, none of these studies investigated how uncertainties in such parameter estimations influence the river depths as well as the flood inundation model outputs, and they have not considered the spatial variability in their analysis."*

155    – Lines 87-89: *"Hence, their results might not be fully representative for such uncertainties in river bathymetry estimations. Also, their research did not consider how these uncertainties affect the flood inundation model outputs."*

-    Lines 90-95: *"Generally, these previous studies have addressed certain gaps in quantifying uncertainties in estimated river bathymetry and show that errors can arise from various sources. However, they have not assessed how the flood inundation model outputs would be affected by errors or uncertainties in the river bathymetry. Additionally, their*
160      *methods did not consider spatial variability in factors used to estimate river bathymetries and their results are not fully representative."*

     In this paper, because we have no information about what the uncertainty is, we look into the sensitivity. Specifically, we developed a sensitivity analysis using Monte Carlo framework which can be used for different formulas and parameters. Within this framework, we chose two formulas that have been validated and used to estimate the river bathymetry at the Waikanae
165 River - the UF and CMR - by Pearson et al. (2023). Due to the time intensity and complexity, we selected only three parameters - river slope, flow, and width - in these formulas and examined how their errors propagate through the flood modelling and affect the outputs. These ideas have been summarised and rewritten for clarity between lines 96-104 as below:

     *"To fill these gaps, we quantified the uncertainty in flood predictions due to errors in the estimated parameters used in two formulas described in Rupp and Smart (2007) and Neal et al. (2021), and validated by Pearson et al. (2023). Within the Monte*
170 *Carlo framework, we generated multiple realisations of river bathymetry, then used them to perform a sensitivity analysis to evaluate the impacts of each parameter on flood predictions, individually and collectively. We also considered the spatial variability in the analysis and whether our number of simulations is large enough to represent our results. This work can contribute to studies of other sources of uncertainty to adequately comprehend the uncertainty in flood model outputs. In the next section, we describe a method to explore relationships between the parameters within those two formulas and show a*
175 *process to examine how errors in these parameters affect the flood predictions."*

     As we do not have information about the sources or uncertainty, based on the observed riverbed elevations, we selected the errors for each parameters from a normal distribution with zero mean and standard deviation set to 10% of the best estimates of parameters. We added this information between lines 190-195 with Figure 3: *"Due to no information about the sources of errors, we assumed that their expected errors would be unbiased and normally distributed with zero mean and a standard*
180 *deviation of 10% of the best-estimated values. This 10% was chosen because: (i) many observed cross-sectional riverbed elevations are within the simulated ensemble range (min-max) of simulated riverbed elevations - calculated from the simulated river bathymetric data (described in detail later in this Section) - as seen in Fig. 3; and (ii) with the same amount of errors, we*

*can then compare the influences of those errors, between datasets, on the flood model outputs.".* This helps us to see how the errors propagate through the flood modelling and affect the flood predictions.

185     Different sources of errors should also be considered, but due to the time intensity and complexity, another research would be a better fit. This information is added in the Discussion Section between lines 419-423: *"Due to the lack of information about the sources of errors, the expected errors in our research were assumed to be unbiased and normally distributed with zero mean and a standard deviation of 10% of the best-estimated values. Hence, different realistic sources of errors should be considered to compare their impacts on the flood predictions. However, owing to the time intensity and complexity, this issue*

190 *should be researched in another study."*

    The main results have shown that between two formulas, the errors in the parameters using the UF formula are associated with greater uncertainty in flood predictions than the CMR formula. Apart from that, when validating the simulations with the observed flood data, the RMSEs between using two formulas are not much different. These key results demonstrate that the flood extent and flood depth are more sensitive to the UF formula than the CMR formula, but this did not translate to

195 substantially reduced RMSEs in the validation. Moreover, the small difference in the RMSEs suggested the applicability of the UF formula to estimate the river without the need of river categorisation. Nevertheless, further research is needed to compare the UF formula with other approaches. This has been shown in Section 3.3, 3.4, and 3.5. It was also mentioned between lines 404-406 and rewritten for clarification: *"... However, because we have only compared the UF formula with the CMR developed for coarse-grained rivers, comparisons with other formulas and approaches are still needed to confirm the applicability of the*

200 *UF formula.".*

    Between the parameters considered in this study, the uncertainty in flood model outputs associated with the river slope parameter is the smallest, followed by the river flow and width. This information can support the data collection process when resources are limited. Specifically, we can focus on measuring the parameters that have the greatest impacts (river flow and width) and deprioritize the ones associated with the lowest influences (river slope). However, as mentioned above, due to the

205 time intensity and complexity, we have not explored the errors in the river Manning's n as well as $\alpha$ and $\beta$ coefficients. This whole information was mentioned between lines 407-415 and was rewritten for clarification as below:

    *"The results of our research can help the data collection process in which the parameters that have the greatest impact (specifically river flow and width) should be focused on measuring if resources are limited. Meanwhile, the parameter associated with the lowest influence (river slope) can be deprioritised. Nevertheless, due to the time-intensity and complexity,*

210 *we have not explored the errors in the river Manning's n as well as $\alpha$ and $\beta$ coefficients. Furthermore, the Waikanae River bank-full flow is not strongly correlated with the variability of the bathymetry along the river as it nearly stays constant. This is based on the fact that the Waikanae River sections in our paper were not joined by major tributaries. Hence, future studies should investigate the errors associated with these factors and perform a thorough sensitivity analysis to better support the data collection process.".*

215     Generally, our focus is to find out how the errors in estimated river bathymetric data can affect the flood inundation model outputs. Since we have no information about what the uncertainty is, we look into the sensitivity. Furthermore, there are many sources of errors for such parameters and they would be different for different formulas. Hence, we developed this sensitivity

analysis using the Monte Carlo framework that can be applicable to various formulas and parameters to assess which parameters that have errors can affect significantly on the flood model outputs. This also supports the data collection process, allowing it to

220 focus on these parameters if the resources are limited and suggests future investigations to research errors in these parameters.

*Question:* Besides, the study didn't use any measurement data to validate the estimated bathymetry, so the analysis actually only shows the range of estimated bathymetry caused by a 10% variation in S/Q/w, which, in my opinion, is a rather direct procedure from the viewpoint of mathematic, since the formulas for bathymetry (Eq.2) is a very simple equation.

The estimated river bathymetry for the Waikanae River was already validated by Pearson et al. (2023) (added at lines

225 97). Apart from that, as the reviewer said, this sensitivity of the 10% change on the equations is simple to compute, but understanding how that then affects the flood model ouputs is not straightforward and that is what this manuscript investigates. In other words, this analysis provided information about how the errors in the estimated river bathymetric data can propagate and affect the flood inundation model outputs.

**2.4    Question 4**

230 *Question:* Some questions about UF and CMR formulas. 1) According to section 2.2 the only difference in these two formulas is the different value of $\alpha$ and $\beta$, am I right? 2) Table 1: In my understanding, Manning's n should be a parameter reflecting the characteristics of riverbed. Why is it different in different formulas?

*Answer:* The two formulas have different values of $\alpha$, $\beta$, and river Manning's n. The Conceptual Multivariate Regression formula has a constant river Manning's n because it is developed specifically for coarse-grained rivers. To highlight this idea,

235 the information between lines 160-162: *"The CMR formula, designed for coarse-grained rivers, was selected to match with Waikanae River (Gyopari et al., 2014), and the UF formula was chosen for its simplicity (Neal et al., 2021) and can be widely applicable.".* Also, the information between lines 166-168 were rewritten: *"For the $\alpha$ and $\beta$ coefficients, the UF formula used constant values of 2/3 and 1/2 respectively, while the CMR formula, designed for coarse-grained rivers, applied 0.745 and 0.305 respectively with a constant value of 0.162 for Manning's n.".*

**240    3    Reviewer 2**

*Summary of reviewer's comments:* The reviewer was asking for further information and clarification about the flood model used in the paper, adequate assessment of the uncertainty in flood model outputs, summary of previous publication - Nguyen et al. (2024b), context of this uncertainty analysis, and more case studies for robustness.

*Summary of authors' responses:* We thank you so much for your help in pointing out where we lack clarification and need

245 improvement. In general, we have added further information about LISFLOOD-FP - the flood model used in the paper. In this response, we have also indicated how we analysed the flood model outputs in the paper and summarised the previous publication - Nguyen et al. (2025). Finally, we have explained the robustness of our paper and suggested that another study would be better to investigate a wide range of rivers.

**3.1 Question 1**

250 **Question:** Generally, the authors need to significantly improve the methods section to convey the methods used in this study. In particular, they did not provide enough explanation regarding the LISFLOOD modeling and the input data utilized. It is important to summarize the processes implemented by the model to understand the relationships presented in the results section. For example, processes such as, backwater effect, sediment processes, human regulations, etc.

*Answer:* The information about LISFLOOD-FP model was rewritten for clarification and added more information between 255 lines 143-155: *"In this study, LISFLOOD-FP (Bates et al., 2010; Neal et al., 2018), a 2D hydrodynamic model, was used to simulate the January-2005 flood event (which was calibrated for this site in Nguyen et al. (2025)) because it is well known for its computational efficiency and highly accurate flood model outputs (Nguyen et al., 2025). The DEM and Manning's n values, along with the flow information and tidal data mentioned above were used as input into this model.*

*In LISFLOOD-FP, the formula to compute the water flow Qcell at the interface $i+1/2$ between cells $i$ and $i+1$ over a time* 260 *step $\Delta t$ is:*

$$Qcell_{i+1/2}^{t+\Delta t} = \frac{q_{i+1/2}^t - gh_{flow}^t \Delta t Scell_{i+1/2}^t}{[1 + \frac{g\Delta t n^2 |q_{i+1/2}^t|}{(h_{flow}^t)^{7/3}}]} \Delta x \tag{3}$$

*where $q^t$ represents the flux at time $t$, $\Delta x$ denotes the cell width, $Scell$ and $h_{flow}$ are the water surface slope and flow depth between cells (Bates et al., 2010). The flow formula here is displayed for the x direction, the y direction can be obtained analogously. The cell water depth $h_{flow}$ is updated based on the discharge through the four boundaries of that cell as below,* 265 *where $i$ and $j$ denote the cell coordinates (Shustikova et al., 2019):*

$$\frac{\Delta h_{flow}^{i,j}}{\Delta t} = \frac{Qcell_x^{i-1,j} - Qcell_x^{i,j} + Qcell_y^{i,j-1} - Qcell_y^{i,j}}{\Delta x^2}." \tag{4}$$

The main inputs for the LISFLOOD-FP model to simulate the January-2005 flood event are the river flow data, tidal data, DEM, and Manning's n converted from roughness length. The main outputs of the model are the water surface elevation and water depth across the time series and their maximum values. Among them, the study chose the maximum water depths 270 (MWDs) and maximum water surface elevations (MWSEs) to analyse. The information about these inputs and outputs was mentioned between lines 217-226:

*"Similar to Durand et al. (2008); Moramarco et al. (2019); Kechnit et al. (2024), and especially Nguyen et al. (2025), our research also applied a Monte Carlo framework to generate 50 DEMs and 50 Manning's n maps from those 50 simulated rivebed elevations and LiDAR data from OpenTopography (2013) using the method described in Section 2.1 for each dataset.* 275 *These 50 DEMs and 50 Manning's n maps are the same except for the river locations due to the use of 50 different simulated riverbed elevations. Hence, we only focus on analysing the variation in the simulated river bathymetric data used to generate these riverbed elevations instead of those simulated topographic data (see Section 3.2.). The DEMs and Manning's n maps that include the simulated river bathymetric data, along with the January-2005 flow and tidal data mentioned in Section 2.1, were*

*then used in the LISFLOOD-FP flood model to produce 50 maximum water depths (MWDs) and 50 maximum water surface*
280 *elevations (MWSEs) for further statistical analysis."*

According to Bates et al. (2010), the LISFLOOD-FP flood model does not assume uniform flow and includes the surface slope, $Scell$, which allows the model to simulate the situations like backwater effects - where the water flows uphill or slows down due to downstream resistance like tides. However, the flood model does not include sediment processes. Also, for human regulations, it partially supports by representing the levees/embankments, for example, through the DEM or manually inserted
285 structures. In this study, the LISFLOOD-FP flood model and our research focuses on the pixel-level hydrodynamics and spatial water depth. Hence, the processes including backwater effect, sediment processes, and human regulations are not our main focus.

**3.2 Question 2**

We divided it into two sub-questions to answer as below:

290 *Question:* I do not believe the authors adequately assess the uncertainty of floods, as they primarily evaluate the uncertainty of DEMs, bathymetry estimations, and roughness coefficients.

*Answer:* The reviewer is correct, we do not assess all the uncertainties in floods as this would be an incredibly big task. Instead, we focus on this one particular aspect of uncertainty - the errors in river bathymetry estimation that can affect the flood predictions - to understand the bigger picture. To highlight this idea better, we have rewritten lines 55-104 as follows:

295 – Lines 55-63: *"Regardless of any approaches to estimate the river bathymetric data, due to the inability to capture the randomness of the real-world river systems, these estimations still contain errors. These errors can cause the simulated river bathymetries to deviate significantly from the actual ones. Consequently, using these modelled river bathymetries to represent the rivers in food inundation modelling can affect the flood predictions. Currently, several studies have investigated the errors in the estimated river bathymetry (Durand et al., 2008; Lee et al., 2018; Moramarco et al.,2019;*
300 *Kechnit et al., 2024), but they have not considered how these estimations with errors affect the flood model outputs."*

– Lines 64-73: *"For instance, Durand et al. (2018) developed an ensemble-based data assimilation approach for estimating river bathymetry from water surface elevation measurements and the LISFLOOD-FP hydrodynamic model. Using a Monte Carlo-based framework, they also performed a sensitivity analysis to assess how various error sources affected the estimated results. Their study found that errors in some input factors for their approach, such as river roughness and*
305 *flow conditions, have greater influence than the water surface elevation measurement errors. However, this research did not evaluate how the errors in these river bathymetric estimations can affect flood model outputs with consideration of spatial variability of input factors in the analysis."*

– Lines 79-83: *"Nevertheless, none of these studies investigated how uncertainties in such parameter estimations influence the river depths as well as the flood inundation model outputs, and they have not considered the spatial variability in*
310 *their analysis."*

- Lines 87-89: *"Hence, their results might not be fully representative for such uncertainties in river bathymetry estimations. Also, their research did not consider how these uncertainties affect the flood inundation model outputs."*

- Lines 90-95: *"Generally, these previous studies have addressed certain gaps in quantifying uncertainties in estimated river bathymetry and show that errors can arise from various sources. However, they have not assessed how the flood inundation model outputs would be affected by errors or uncertainty in the river bathymetry. Additionally, their methods did not consider spatial variability in factors used to estimate river bathymetries and their results are not fully representative."*

- Lines 96-104: *"To fill those gaps, we quantified the uncertainty in flood predictions due to errors in estimated parameters used in two formulas described in Rupp and Smart (2007) and Neal et al. (2021), and validated by Pearson et al. (2023). Within the Monte Carlo framework, we generated multiple realisations of river bathymetry, then used them to perform a sensitivity analysis to evaluate the impacts of each parameter on flood predictions, individually and collectively. We also considered the spatial variability in the analysis and whether our number of simulations is large enough to represent our results. This work can contribute to studies of other sources of uncertainty to adequately comprehend the uncertainty in flood model outputs. In the next section, we describe a method to explore relationships between parameters within those two formulas and show a process to examine how errors in these parameters affect the flood predictions."*

Now, we briefly explain how we analyse this uncertainty in flood predictions arising from the estimated river bathymetry. The variability of simulations of the topographic data like DEM and roughness length/Manning's n gathers around the river due to the use of simulated river bathymetric data. Hence, we only focus on the variability of the simulated river bathymetric data (mentioned at lines 220-223) as analysed in Figures 5 and 6 in the paper.

After that, we analysed how the variability in the river bathymetric data can affect the flood model outputs based on MWDs and MWSEs. Specifically, for each of eight datasets of 50 MWDs, we computed their mean (mMWDs), standard deviation (sdMWDs), and coefficient of variation (covMWDs). Here, because the mMWDs and sdMWDs did not provide further insights, they are not considered in the paper. We also calculated proportion of simulations in which a given pixel was flooded (pFs) to distinguish where was always flooded, never flooded, and sometimes flooded throughout these realisations. These ideas were mentioned between lines 217-237.

For the flood extent, its calculation information was added between lines 234-237: *"Additionally, we computed expected flooded area or expected flood extent, a metric often employed by decision-makers, for each simulation for comparison. The expected flood extents were calculated based on these pFs by multiplying the area of one pixel (10 m x 10 m) with number of pixels that were always and sometimes flooded."*. Apart from this, we also validated each flood simulation - MWSE with the observed flood data - flood levels using the RMSE metric. This information was mentioned between lines 240-243: *"In our research, we went further than Nguyen et al. (2025) by validating each flood simulation - MWSE with the observed flood levels measured by Wallace (2010) for the January-2005 event. The Root Mean Square Error (RMSE) metric was harnessed for these validations."*

The results of covMWDs, flood extents, and RMSEs were shown in Figures 7-8, Figures 9-10, and Figure 11. This corresponds to the Sections 3.3, 3.4, and 3.5. In particular, we used boxplots (Figures 7, 9, 12) to compare the variations of eight datasets and maps (Figures 8 and 20) to visualise and explain. Here, our explanations linked with what we found in the variations of simulated river bathymetric data as shown in Figure 5 and mentioned in Section 3.2.

Generally, based on those Figures, we found that the variations in the MWDs based on covMWDs and flood extents correspond to the variations in the estimated river bathymetries. In particular, with the same amount of uncertainty added to each of eight datasets, the variation in the slope parameter corresponds to the smallest variation in the MWDs, followed by the flow and the width. Between two formulas, the errors in the parameters of the UF formula are associated with greater uncertainty in the MWDs than those of the CMR formula. We provided the explanation as below for each Section:

– For covMWDs, Section 3.3, lines 354-359: *"To explain, between parameter datasets, the small variability in the river bathymetry corresponding with the variation in the river slope does not significantly affect the water spreading into the floodplain, unlike the variations in the river bank-full flow and width. The impacts of all these variations become more apparent in floodplains farther from the river, especially at flood boundaries in midstream, where the water has less direct connection with the river. Between two formulas, because the variations in the UF-formula river bathymetries are higher than the CMR-formula ones as seen in Fig. 5, the variations in the flood depths of the UF-formula datasets are also higher than the CMR-formula datasets."*.

– For flood extents, Section 3.4, lines 371-376: *"Between the two formulas, the blue zoomed-in images highlight a location surrounding the river upstream to 1000 m downstream where the UF-formula river bathymetries are lower than the CMR-formula ones, resulting in greater flood extent here in the UF-formula datasets. This leads to that, in the UF-formula datasets, the flood extent variation appears not only in locations already totally flooded in the CMR-formula but also in new regions that are never flooded in the CMR-formula datasets. Consequently, there are more variations in flood extent in the UF-formula datasets compared to the CMR-formula datasets."*.

– For RMSEs, Section 3.5, lines 385-389: *"To explain, the CMR is developed for coarse-grained rivers like the Waikanae River, leading to lower RMSEs than the UF formula. In contrast, the UF formula was not developed for any specific river types, which may contribute to its slightly higher RMSE. However, these small differences in RMSEs between the two formulas highlight a broad applicability of the UF formula on rivers without categorising their types."*.

*Question:* In addition, the authors did not indicate whether their sources of uncertainty are valid by referring to the ranges of DEM values, any reported roughness, etc.

*Answer:* As we do not have information about the sources of uncertainty, based on the observed riverbed elevations, we selected the errors for each parameters from a normal distribution with zero mean and standard deviation set to 10% of the best estimates of that parameter. We added this information between lines 190-195 with Figure 3: *"Due to no information about the sources of errors, we assumed that their expected errors would be unbiased and normally distributed with zero mean and a standard deviation of 10% of the best-estimated values. This 10% was chosen because: (i) many observed cross-sectional*

*riverbed elevations are within the simulated ensemble range (min-max) of simulated riverbed elevations - calculated from the simulated river bathymetric data (described in detail later in this Section) - as seen in Fig. 3; and (ii) with the same amount of errors, we can then compare the influences of those errors, between datasets, on the flood model outputs.*

380    Different realistic sources of errors should also be considered, but due to the time intensity and complexity, another research would be a better fit. This information is added in the Discussion Section between lines 419-423: *"Due to the lack of information about the sources of errors, the expected errors in our research were assumed to be unbiased and normally distributed with zero mean and a standard deviation of 10% of the best-estimated values. Hence, different realistic sources of errors should be considered to compare their impacts on the flood predictions. However, owing to the time intensity and complexity, this issue*
385    *should be researched in another study."*

**3.3    Question 3**

***Question:*** In addition, they refer to a previous publication, Nguyen et al. (2024b), to get the key details for the methods used. For a smooth reading experience, the authors should summarize that key information in this manuscript as well.

*Answer:* A summary of the key details of the methodology from Nguyen et al. (2025) pertinent to this work was added
390    between lines 109-119 of Section 2: *"Our data and methodology were based on Nguyen et al. (2025) where the uncertainty in flood predictions due to arbitrary conventions in grid alignment was quantified. To explain, their research is also about how the uncertainty in the process of generating the topographic data like DEM and roughness length can propagate through the flood modelling to the outputs. Hence, their data and methodology can be applied in our research.*.

*Accordingly, we simulated the same flood event using the LISFLOOD-FP flood model and applied a similar method to*
395    *generate topographic data. Moreover, a Monte Carlo framework was also designed in our research to observe how the uncertainty in estimated river bathymetries propagates through the flood modelling to the outputs. To assess the uncertainty, some similar measurements were used, some were not because they did not provide further information, and some were added to understand better the uncertainty. These similarities will be mentioned in details in the sections below."*.

**3.4    Question 4**

400    ***Question:*** Moreover, the authors did not explain the context of this uncertainty analysis of the flood prediction. Also, they need to include the details about the flood event they used in this study to demonstrate what they want to establish from this study.

*Answer:* The context of this uncertainty analysis of the flood was explained and rewritten for clarification as below:

– Lines 20-27: *"River bathymetry refers to the river depth measurement (Panigrahi, 20140). It plays a crucial role in flood modelling because it determines when and where water leaves the river channel and starts to flood overland (Cook*
405    *and Merwade, 2009; Awadallah et al., 2022). Currently, hydrographic surveys and remote sensing methods, especially swath beam sonar and blue-green LiDAR, are prevalently employed to obtain these river bathymetric data (Coasta et al., 2009; Kinzel et al., 2013; Dey et al., 2019). Multi-beam sonar is effective but time-consuming, while blue-green LiDAR is faster but does not work in sediment-laden or deep water, and both of them are expensive (Bailly et al., 2010;*

*Flener et al., 2012; Bures et al., 2019). For these reasons, various approaches have been proposed to estimate these data*
*(Ghorbanidehno et al., 2021; Araujo and Hedley, 2023).".* We have changed "... unable to obtain measurements ..." to
"... does not work ..." and added "... and both of them are expensive ...".

- Lines 55-63: *"Regardless of any approaches to estimate the river bathymetric data, due to the inability to capture the randomness of the real-world river systems, these estimations still contain errors. These errors can cause the simulated river bathymetries to deviate significantly from the actual ones. Consequently, using these modelled river bathymetries to represent the rivers in food inundation modelling can affect the flood predictions. Currently, several studies have investigated the errors in the estimated river bathymetry (Durand et al., 2008; Lee et al., 2018; Moramarco et al.,2019; Kechnit et al., 2024), but they have not considered how these estimations with errors affect the flood model outputs."*

- Lines 90-95: *"Generally, these previous studies have addressed certain gaps in quantifying uncertainties in estimated river bathymetry and show that errors can arise from various sources. However, they have not assessed how the flood inundation model outputs would be affected by errors or uncertainties in the river bathymetry. Additionally, their methods did not consider spatial variability in factors used to estimate river bathymetries and their results are not fully representative."*

- Lines 96-104: *"To fill those gaps, we quantified the uncertainty in flood predictions due to errors in estimated parameters used in two formulas described in Rupp and Smart (2007) and Neal et al. (2021), and validated by Pearson et al. (2023). Within the Monte Carlo framework, we generated multiple realisations of river bathymetry, then used them to perform a sensitivity analysis to evaluate the impacts of each parameter on flood predictions, individually and collectively. We also considered the spatial variability in the analysis and whether our number of simulations is large enough to represent our results. This work can contribute to studies of other sources of uncertainty to adequately comprehend the uncertainty in flood model outputs. In the next section, we describe a method to explore relationships between parameters within those two formulas and show a process to examine how errors in these parameters affect the flood predictions."*

We rewrote and added more information about the study site as well as the flood event between lines 121-130: *"Similar to Nguyen et al. (2025), the Waikanae River, located on the West Coast of the Wellington Region in New Zealand, was used in this paper. Its catchment covers around 149 km$^2$ and spans from the Tararua Ranges to the West Coast. There are recurring flooding issues at this study site that have influenced the regions around the river..*

*In this study, we simulated a flood event with an 80-year return period that occurred in Waikanae from January $5^{th}$ to $7^{th}$, 2005 and reached its peak on 6th. Here, we focused on fluvial flooding from the Waikanae River. This allowed us to observe how the uncertainty in the estimated river bathymetric data can impact the flood inundation model outputs. Figure 1a depicts our site study extending about 7 km from the Waikanae Water Treatment Plant gauge to the coast. Figures 1b and 1c show the flow information recorded at the gauge by the Greater Wellington Regional Council (2005) and the tidal data estimated by the NIWA Tide Forecaster (2005) respectively.".*

**3.5 Question 5**

***Question:*** The authors need to enhance their experimental methods to strengthen the robustness of their findings, such as applying these analyses to various case studies across a wide range of rivers.

*Answer:* As we mentioned in the paper and in question 4, there are many approaches to estimate the river bathymetric data. However, due to the inability to capture the randomness of the river systems, errors in the estimations can introduce uncertainties that significantly deviate the simulated river bathymetries from the actual ones. Consequently, using these estimated river bathymetries to represent the river in the flood modelling can affect the flood predictions. Previous studies have investigated such errors in the river bathymetry estimations, but they did not evaluate how the flood model outputs would be affected if those estimations were used to represent the rivers in the flood modelling.

To contribute to this field, we quantify the uncertainty in flood predictions due to the estimated river bathymetries. In our case, we have investigated how the errors inherent in the estimated parameters (river slope, flow, and width) used in two formulas (CMR and UF) can propagate through the flood modelling to the flood predictions. These formulas were validated by Rupp and Smart (2007) and Neal et al. (2021), and validated for the Waikanae River and Buller River by Pearson et al. (2023).

Although our research was conducted at one study site, it helps raise the awareness of flood modellers who use estimated river bathymetries in their flood modelling. Furthermore, in this study, we provided a thorough Monte Carlo framework to capture the uncertainty in the flood model predictions arising from the estimated river bathymetry. We designed this framework including spatial variability which was not considered by previous studies (Durand et al., 2008; Lee et al., 2018; Moramarco et al., 2019; Kechnit et al., 2024) and using larger number of simulations than theirs. We also performed a sensitivity analysis collectively and individually. Hence, this framework can be applied to a wide range of formulas used to estimate river bathymetries.

Between lines 391-400, we rewrote for clarification: *"Our research went a step further than previous studies (Durand et al., 2008; Lee et al., 2018; Moramarco et al., 2019; Kechnit et al., 2024) to quantify the uncertainty in flood predictions due to the errors in the estimated river bathymetry. It helps raise the awareness of flood modellers who also use estimated river bathymetries in flood modelling. In this research, we applied the Monte Carlo method to generate a large number of simulations to capture the typical variability in the flood predictions and included spatial variability in our method. Moreover, we not only considered associated error distributions in parameters collectively, but we also performed a sensitivity analysis to assess the impact of each parameter. This analysis framework can then be applied to a wide range of formulas that are used to estimate river bathymetries to represent rivers in the flood modelling."*.

Based on our results, we found out some key points that can develop further research. Specifically, we have suggested the applicability of the UF formula without the need of river categorisation. However, we have only compared it with the CMR formula and this still needs further investigations with other equations. Between lines 404-406, we rewrote the idea for clarification: *"... However, because we have only compared the UF formula with the CMR developed for coarse-grained rivers, comparisons with other formulas and approaches are still needed to confirm the applicability of the UF formula."*.

We also observed that the uncertainty in flood predictions associated with the errors in the river slope parameter is the smallest, followed by the river flow, and width. This information can help the data collection process in which the parameters

475 that have the greatest impact (specifically flow and width) should be focused on measuring if resources are limited. Meanwhile, the parameter associated with the lowest influence (river slope) can be deprioritised. However, due to the time-intensity and complexity, we have not explored the errors in the river Manning's n as well as $\alpha$ and $\beta$ coefficients. Hence, further research is necessary to perform a more thorough sensitivity analysis between these parameters and perhaps between formulas.

The above ideas were rewritten between lines 407-415: *"The results of our research can help the data collection process*
480 *in which the parameters that have the greatest impact (specifically river flow and width) should be focused on measuring if resources are limited. Meanwhile, the parameter associated with the lowest influence (river slope) can be deprioritised. Nevertheless, due to the time-intensity and complexity, we have not explored the errors in the river Manning's n as well as $\alpha$ and $\beta$ coefficients. Furthermore, the Waikanae River bank-full flow is not strongly correlated with the variability of the bathymetry along the river as it nearly stays constant. This is based on the fact that the Waikanae River sections in our paper*
485 *were not joined by major tributaries. Hence, future studies should investigate the errors associated with these factors and perform a thorough sensitivity analysis to better support the data collection process."*.

In practice, different rivers will have different characteristics, so we agree that the suggestion for applying our investigation to various case studies across a wide range of rivers is necessary. However, due to the amount of work, we leave this to future research. We have added this idea after line 416-418: *"In practice, different rivers will have different characteristics. Hence,*
490 *it is necessary to generalise this study by considering a wide range of rivers for comparison and confirmation for the results found here. Accordingly, further research focusing on many rivers with diverse features is recommended."*.

**4 Extra changes**

**4.1 Rewrite abstract**

We rewrote the abstract to enhance the readability and consistency and also for clarification at lines 1-18 as below. In that the
495 previous information - *"The results indicate that, between the two methods, the combined errors in the parameters using the Uniform Flow formula are associated with greater uncertainty in flood depths (median error: 3.89 m, quartile range: 2.36 to 7.78 m) and extents (208.72 ha, 206.59 to 209.58 ha), compared to Conceptual Multivariate Regression (depth: 3.61 m, 2.32 to 7.37 m; extent: 207.82 ha, 206.42 to 208.48 ha)"* - was removed because this belongs to intial analysis and we decided not to use this information anymore.

500 **Abstract:** *"River bathymetry is important for accurate flood inundation modelling but is often unavailable due to the time-intensive and expensive nature of its acquisition. This leads to several proposed and implemented approaches for its estimation. However, the errors in estimations inherent in these methods and how they affect the accuracy of the flood inundation modelling outputs, has not been extensively researched. Hence, to contribute, we investigate the sensitivity of flood predictions to the errors in river slope, width, and bank-full flow used in two formulas - the Uniform Flow and the Conceptual Multivariate Regression*
505 *- for estimating river bathymetry. In this study, we employed a Monte Carlo framework to introduce random errors into these parameters drawn from a normal distribution with zero mean and a standard deviation set to 10% of their best estimates. Using this process, we generated 50 simulated river bathymetries for each parameter along with an additional 50 where the*

*errors were applied to all parameters simultaneously. The riverbeds generated from these bathymetries were combined with topographic LiDAR data to create model grids. Each grid was used in the hydrodynamic model LISFLOOD-FP to simulate the 2005 flood event in the Waikanae River area of New Zealand. We assessed the resulting flood inundation predictions for their variability and sensitivity. The results indicate that between two methods, the errors in the parameters in the Uniform Flow formula are associated with greater uncertainty in flood inundation depths and extents compared to the Conceptual Multivariate Regression. Among the parameters, the width errors correspond to the highest uncertainty, while the slope errors correspond to the lowest.".*

**4.2 Rewrite Section 2.1. Study site and data source**

In Section 2.1., we also rewrote lines 131-139 for clarification: *"Following the approach of Nguyen et al. (2025), the topographic data - DEM and roughness length - in our paper were generated by an open-source Python package, GeoFabrics (version 0.9.4) developed by Pearson et al. (2023). Specifically, the package sampled and interpolated LiDAR point cloud data downloaded from OpenTopography (2013) onto a 10-metre square grid using Inverse Distance Weighted – an interpolation method has been commonly used in flood modelling (Ibrahim and Fritsch, 2022; Xing et al., 2022; Huang et al., 2023). To represent the river in this process, since the LiDAR only contains the water surface elevations, the estimated riverbed elevation data were then obtained to be included in the point cloud data by subtracting the estimated river bathymetric data or river depths (see Section 2.2) from these water surface elevations. The roughness length was converted to Manning's n using a conversion developed by Smart (2018)"*

**4.3 Rewrite Conclusion**

In the Conclusion Section, we rewrote between lines 441-457 to match with what we adjusted in the Discussion Section as below:

- Lines 451-461: *"Our analysis framework can be applied to various formulas used to estimate the river bathymetries to represent rivers in the flood modelling. The slight differences in RMSEs between the two formulas suggest a broad applicability of the UF formula across many river types without categorising them, but further study is still necessary to confirm this. Moreover, our results can support the data collection process by directing it to focus on measuring the parameters that have more significant impacts on the flood inundation model outputs if the resources are limited. Additionally, due to the time-intensity and complexity, the river Manning's n, and $\alpha$ and $\beta$ coefficients were not considered in our study, and thus further research about these parameters including a thorough sensitivity analysis are recommended."*

- Lines 462-468: *"Apart from that, since different rivers have different characteristics and our work only focuses on the Waikanae River, another study implemented on many rivers with different features is essential. In addition, further investigation should consider how different realistic sources of errors affect the flood predictions. Another future topic of interest is the impact of grid resolution on the estimated river bathymetry, which then influences the flood inundation*

*predictions. Currently, to cover such uncertainty, a freeboard is often used, but it fails to cover the variation in the flood*
*extent, and thus a further study is needed to improve its effectiveness. Lastly, there is a need for simpler and faster method*
*than Monte Carlo framework such as machine learning approaches."*

**4.4 Correct some grammars and vocabularies**

There are some grammar and vocabulary mistakes we would like to edit for accuracy and consistency as below:

– Changed "categorize" to "categorise" at lines 28, 40, 389, 404, and 453.

– Changed "minimize" to "minimise" at line 76.

– Changed "visualize" to "visualise" at lines 181, 238, 244, 364.

– Rewrite lines 215-216: "Eight datasets of these simulated river data were organised and presented in the Table 2

**4.5 Changing styles of Figures and Tables**

We re-styled all the Figures and Tables for readibility as below:

– Figure 1: We combined the hydrograph and tidal graph into one (b) and changed the caption: *"Study site and data source*
*(adapted from Nguyen et al., 2025): (a) Waikanae River flow discharge recorded by the (Greater Wellington Regional*
*Council, 2005) and tidal data recorded by the (NIWA Tide Forecaster, 2005) for the flood event from $5^{th}$ to $7^{th}$ January,*
*2005."*

– Figure 8 and 10: We combined all subfigures into one figure.

– Table 1 and 2: We created these tables directly in Latex rather than added them as pictures.

---

## Author Response (AR2)

**Responses to reviewers on: "Quantifying uncertainty in flood predictions due to river bathymetry estimation" - $2^{nd}$ round**

Martin Nguyen[1, 2, 3], Matthew D. Wilson[1, 3], Emily M. Lane[4], James Brasington[2, 3], and Rose A. Pearson[4]

[1]Geospatial Research Institute, University of Canterbury, Christchurch, New Zealand
[2]Waterways Centre, University of Canterbury, Christchurch, New Zealand
[3]School of Earth and Environment, University of Canterbury
[4]National Institute of Water and Atmospheric Research (NIWA), Christchurch, New Zealand

**Correspondence:** Martin Nguyen (tmn52@uclive.ac.nz)

Dear Prof. Lixin Wang and reviewers,

We thank you so much for the second opportunity to respond to the reviewers on "Quantifying uncertainty in flood predictions due to river bathymetry estimation". We really appreciate the time and effort that you and the reviewers spent on providing valuable recommendations to our manuscript. Similarly to the first round, please see below, in green, for our responses to the reviewers' questions. We separated the responses into three sections for the editor and two reviewers, with an extra section to list all parts we edited to make the content more accurate, consistent, and concise. The sections and lines mentioned here are based on the track-changed manuscript version.

**1  Editor report**

Both reviewers think the manuscript was significantly improved after the revision. However, both reviewers think the manuscript requires further improvements. For example, as one reviewer pointed out, there are still opportunities to improve its clarity, justify the assumptions, validate and generalize the findings, and elaborate on the limitations. I generally concur with the reviewers' assessment and recommend a modest revision of the manuscript.

We would like to thank the editor again for the second chance to revise our manuscript according to suggestions from the reviewers. In general, in this revision, we have improved the clarity of the uncertainty propagation process, provide further explanation on calibrations and observations, improve the visualisation, elaborate on the limitations, and enhance the result analysis. These will be presented in this response as well as the revised manuscript.

**2  Reviewer 1**

***Summary of reviewer's comments:*** The reviewer recommended further explanation on the uncertainty propagation through the flood model in the manuscript.

***Summary of authors' responses:*** We would like to thank you very much for showing us that we lack this information in the manuscript. We have added further explanation about this uncertainty propagation process into the revised manuscript. This

addition can be seen clearly in the track-changed version in Section 2.2. at lines 143-153 and it is also be explained in detail below in Section 2.1. (question 1) in this response.

**2.1 Question 1**

25 *Question:* I would like to emphasize that the flood model and its performance require a more detailed description. This component is critical, as it links the uncertainties in topographic data to those in flood predictions. As noted in the first-round revision and acknowledged by the authors, quantifying uncertainties in topographic estimation on how uncertainties in topographic estimation propagate through to the flood simulations. Therefore, the manuscript should include more information on how uncertainties in topographic estimation propagate through to the flood simulations.

30 *Answer:* To add further information about the uncertainty propagation through the flood model, first, we moved the flood model description into a separate section at line 129 and named "2.2. Flood model and explanation about uncertainty propagation process". We then added the explanation about the uncertainty propagation process below at lines 143-153:

*"To further expand on the description of uncertainty propagation through the model given in Section 1, we apply the following chain for easier comprehension:*

35 *Estimated river bathymetric data $\longrightarrow$ riverbed elevations $\longrightarrow$ topographic data (DEM and Manning's n derived from roughness length) generated by riverbed elevations and LiDAR data $\longrightarrow$ inputs to a flood inundation model (LISFLOOD-FP in this study) $\longrightarrow$ affects flood model outputs (extents, depths, etc.)*

*As indicated in the chain above, the estimated river bathymetric data that contain errors are used to calculate the riverbed elevations (see Section 2.1). These riverbed elevations are then used to represent the river in the topographic data. These*
40 *topographic data are then inputted into the flood model as a discretisation of the floodplain and channel topography to model the water flow. Here, in the flood model, the river as represented in the topographic data controls when, where, and how much the water leaves the channel and starts to flood. Hence, the flood model outputs such as the flood extents and flood depths are affected by how the river is represented. In the next section, we will describe how the river bathymetric data are estimated."*

After that, we rewrote the first paragraph of Section 2 at lines 92-98 to correctly outline this section: *"In this section, we*
45 *first introduce the study site, necessary data, flood model, and explain the uncertainty propagation process. Next, we define two formulas used for river bathymetry estimation and describe a method to explore the relationships between parameters and river bathymetry from these two equations. We then show how to examine these relationships based on the river of the study site. Finally, we design a sensitivity analysis workflow to quantify the uncertainty in the flood model outputs due to errors in the river bathymetry estimations."*

50 ## 3 Reviewer 2

*Summary of reviewer's comments:* The reviewer recommended to improve the manuscript's clarity, justify the assumptions, validate and generalise the findings, and elaborate on the limitations. Additionally, the authors should further refine the figure

and table captions to accurately describe the content of each figure or table. Apart from that, the flow and readability of the manuscript should be enhanced to make it more suitable for publication.

*Summary of authors' responses:* We would like to thank you very much for showing us where we should provide further clarification, explanation, and improve the visualisations. Generally, we have added further information on the selections of formulas, observations, and explained about the calibration. Additionally, we have enhanced the result analysis along with the visualisations and clarified our limitations. Also, we have improved the flow and readability of the manuscript to highlight our findings and suggestions for future research.

**3.1   Question 1**

*Question:* L178 ends without properly concluding the outcome of the analysis.

*Answer:* We have rewritten and added more information for clarity at lines 199-203: *"... Although we do not know what the true errors are in these parameters, these assumed but reasonable ones from the Monte Carlo framework can still meaningfully indicate how the estimated bathymetric data can affect the flood model outputs. In future research, the measured errors can apply the framework already built in this study to compare and confirm the results.*

*Within the process of generating the simulated errors for each parameter, we spatially model the variation of the errors along the river with a Gaussian variogram ..."*

**3.2   Question 2**

We would like to divide this question into two sub-questions to answer as below:

*Question:* Why do authors choose Uniform Flow and Conceptual Multivariate Regression only to represent river channel depth, while most models use a simpler power law equation with bankfull discharge as an independent variable to estimate it?

*Answer:* The Conceptual Multivariate Regression (CMR) was chosen as it is developed for coarse-grained rivers - a common river type in New Zealand like the Waikane River. Additionally, since we would like to compare this equation with a more widely applicable formula that does not require river categorisation but with similar use of parameters, the Uniform Flow (UF) was selected. These rationales were mentioned and rewritten at lines 158-160: *"The CMR formula, designed for coarse-grained rivers, was selected to match with Waikanae River (Gyopari et al., 2014), and the UF formula was chosen for its similar parameters and can be widely applicable."*. Moreover, Pearson et al. (2023) has validated these formulas for the Waikanae River (as mentioned at lines 83-84) and shown that they did a reasonable job of estimating the river bathymetries.

A simpler power law equation with bankfull discharge can be a good candidate to estimate the river bathymetry. However, it would not adequately capture the complexity of the river bathymetries. In addition, the uncertainty in the flood model outputs can be overlooked due to not considering the other factors such as channel slope and width. These parameters also affect the estimated river bathymetries which will then influence the flood model outputs.

Furthermore, using only one parameter like the bankfull discharge does not seem to be adequate to develop a well-representative sensitivity analysis as well as a framework to quantify the uncertainty in flood model outputs. Instead,

85  using formulas with more important parameters (channel slope, flow, width, etc.) which contribute to the estimated river bathymetries, like the CMR and UF, can help us to build up a more representative and widely applicable framework. This framework can then be applied not only back to simpler power law equations but also other formulas.

*Question:* It might be better to include other estimation methods to improve the robustness of the manuscript.

*Answer:* The main focus of this paper is to understand how the errors in the estimated river bathymetric data affect the
90  flood model outputs rather than comparing every possible estimation method. Apart from that, different methods will have different error structures, and it is necessary to account for these errors in a sensitivity analysis. However, due to the lack of a framework for such sensitivity analysis, this paper then developed one that can be widely applied. Hence, in future research, different methods can then use this framework without building up other ones from scratch. This information has already been mentioned and rewritten at lines 418-421: *"Furthermore, the analysis framework in this research can be applied to a*
95  *wide range of formula, such as those that also consider the sediment impacts, that are used to estimate river bathymetries to represent rivers in flood modelling. Future research about this can help to answer which formula constributes the most to the uncertainty in flood model outputs."*

**3.3  Question 3**

*Question:* L225: The authors should explain the observations they have used in this study.
100  *Answer:* Based on this question and Section 3.15. (question 15) in this response, we have rewritten and added further information to explain the observations we used in this study at lines 245-251: *"In our research, we went further than Nguyen et al. (2025) by validating each flood simulation - MWSE with the observed data. Due to the lack of a thorough map of measured flood levels or satellite-based water surface elevations, we used the observed flood levels under point format provided by Wallace (2010). The Root Mean Square Error (RMSE) metric was harnessed for these validations. Locations of the observed*
105  *data where the flood model predicted to be dry across all the simulations were removed to ensure the RMSE focuses only on predicted flooded regions and to avoid skewing the RMSE. We then visualised the distribution of RMSEs across simulations through side-by-side boxplot for comparison."*.

**3.4  Question 4**

*Question:* Why do Nguyen et al. (2025) exclude bathymetry from their calibration? What are the challenges of using river
110  bathymetry as a calibration parameter, too?

*Answer:* Since the calibration using the river bathymetry was not the focus of Nguyen et al. (2025), it was not selected. Furthermore, this process can be complicated and requires being very careful as the river bathymetry can strongly affect when and where the water leaves the river to flood. Specifically, we initially need a good estimate, then we can decide how to alter the river bathymetric data. From here, there would be many degrees of freedom for the alteration (e.g. changing depth at many
115  locations along the river and adjusting channel shape), which adds many unnecessary extra tasks for Nguyen et al. (2025). Hence, the calibration with river bathymetry was not chosen in their paper.

**3.5 Question 5**

*Question:* The presentation of Table 1 is a little strange. What do the "Exponents" parameters for slope, flow, and so on mean? They vary slightly from alpha and beta.

*Answer:* We thank you so much for showing us that we lack information about this. These "exponents" are exponents of each parameter after being processed from the original $\alpha$ and $\beta$. We have added in Appendix A (lines 495-502) more information about this: *"The exponents mentioned in Section 2.2. are exponents of each parameter after being processed from the original $\alpha$ and $\beta$. Specifically, for the UF formula, with $\alpha = 2/3$ and $\beta = 1/2$, it can be processed as below:*

$$h = \left(\frac{nQ}{wS^{1/2}}\right)^{\frac{1}{1+2/3}} \Leftrightarrow h = \left(\frac{nQ}{wS^{0.5}}\right)^{0.6} \Leftrightarrow h = \left(\frac{n^{0.6}Q^{0.6}}{w^{0.6}S^{0.3}}\right) \tag{1}$$

*Hence, the exponents of the slope (S), bankfull flow (Q), and width (w) for the UF formula are 0.3, 0.6, and 0.6. For the CMR formula, with $\alpha = 0.745$ and $\beta = 0.305$, it can be changed as below:*

$$h = \left(\frac{nQ}{wS^{0.305}}\right)^{\frac{1}{1+0.745}} \Leftrightarrow h = \left(\frac{nQ}{wS^{0.305}}\right)^{0.573} \Leftrightarrow h = \left(\frac{n^{0.573}Q^{0.573}}{w^{0.573}S^{0.175}}\right) \tag{2}$$

*Hence, the exponents of the slope (S), bankfull flow (Q), and width (w) for the CMR formula are 0.175, 0.573, and 0.573.".* Apart from that, we added a note *"(see Appendix A)"* into the caption of Table 1: *"The exponents of parameters in the Conceptual Multivariate Regression and Uniform Flow formulas (see Appendix A), and the value ranges (minimum, maximum, and mean) of paramters along the Waikanae River - the river, slope, bank-full flow, width, and Manning's n - used to explore their relationships with the river bathymetry in both formulas.".* Also, we rewrote at lines 165-167: *"The exponents and value ranges of each parameter are shown in Table 1 and some of them are explained in Appendix A."*

**3.6 Question 6**

*Question:* Section 2.3: The authors should include their mini-analysis as supplementary material.

*Answer:* The section now becomes Section 2.4. and we already provided the analysis in Section 3.2. However, to enhance the clarity, we added more information at lines 186-187: *"... Three scatter plots depict the relationship between the variance of each parameter and these combined river bathymetries. All of these visualisations and analysis are provided in Section 3.2. In the next section, we detail how to generate these simulated parameters and corresponding river bathymetries and examine their variations on flood predictions.".* We placed the result analysis as Section 3.2. in the manuscript to later compare with the results from the Monte Carlo framework in Section 3.3. This would help the whole analysis easier to follow and comprehend.

**3.7 Question 7**

*Question:* What is the x-axis "distance" reference point in Figure 3? Between UF and CMR, as well as between estimated and simulated, there is no discernible difference. Perhaps displaying a zoomed-up area would improve visualization. Also, the authors should improve the figure caption to reflect what is presented. What is shown by the color range is not clear.

*Answer:* The x axis "distance" reference point in Figure 3 is the river mouth. In other word, the x axis "distance" is the distance upstream of river mouth. We have added zoomed-in images and increased the line width of simulations to enhance

the visualisation. We have rewritten the legends, figure captions, and added more information about the color range for clarity, easy comprehension, and reflect what is represented as below:

150     – We changed the x axis label *"Distance (m)"* to *"Distance upstream of river mouth (m)"*

    – We changed the legend *"UF - Estimated bed from GeoFabrics"* to *"UF - Estimated riverbed elevations from GeoFabrics"*. Similarly, we changed the legend *"CMR - Estimated bed from GeoFabrics"* to *"CMR - Estimated riverbed elevations from GeoFabrics"*.

    – We changed the legend *"UF - Simulated bed elevations"* to *"UF - Multiple simulated riverbed elevations"*. Similarly, we

155     changed the legend *"CMR - Simulated bed elevations"* to *"CMR - Multiple simulated riverbed elevations"*.

    – We changed the legend *"Observed bed elevation"* to *"Observed riverbed elevations"*.

    – We rewrote the caption and added more information: *"Observed cross-sectional, best estimated (from GeoFabrics of Pearson et al. (2023)), and simulated riverbed elevations at the Waikanae River. The best estimates and simulations of riverbed elevations computed using the Uniform Flow formula are in the first column: (a) slope, (c) bank-full flow, (e)*

160     *width, and (g) combined. The ones calculated using the Conceptual Multivariate Regression formula are in the second column: (b) slope, (d) bank-full flow, (f) width, and (h) combined. The color shading represents multiple simulated riverbed elevations (span of simulations)"*.

    – We also changed "Distance between downstream and upstream (m)" to "Distance upstream of river mouth (m)" for x axis of Figure 5 for consistency.

165 ### 3.8    Question 8

*Question:* Section 2.4 should at least be divided into two sections: one for the evaluation process and another for the Monte Carlo simulation process.

    *Answer:* We thank you very much for this suggestion. We changed the title of Section 2.5 (was Section 2.4) from *"Monte Carlo simulation process"* into *"Monte Carlo framework"* and added more information at lines 189-190: *"Figure 2 shows a*

170 *Monte Carlo simulation process undertaken in this study. To describe the framework in this figure, we divide this section into two subsections. The first is about simulation process and the second is about statistical analysis."*. We then divided this section into two parts as below:

    – *"Section 2.5.1. Simulation process"* from line 191 to line 230.

    – *"Section 2.5.2. Statistical analysis"* from line 231 to line 251.

**3.9 Question 9**

*Question:* L236-237: But "steeper rivers" also erode more sediment. L240: Larger, wider rivers deposit more sediment into the riverbed. However, this type of phenomenon varies over time and is not represented by the model used by the authors.

*Answer:* The formulas that consider sediment dynamics often require additional data that might not be available or beyond the scope of this study (e.g. bed composition and sediment load). Hence, in this paper, for simplicity and to focus on how the uncertainty in the river bathymetric data impacts on the flood model outputs, we selected the two hydraulic equations - the Uniform Flow and Conceptual Multivariate Regression - that do not consider the sediment effects. We have already mentioned this at lines 295-297: *"The above findings are based on the variation in the river bathymetry when a parameter is changed while others remain constant. Also, we have not considered other factors such as sediment load in this analysis. Hence, these results should not be used to fully reflect the real-world river systems."*. However, to enhance the clarity, we rewrote the lines the reviewer mentioned as below:

- Lines 259-262 (was lines 236-237): *"Physically, when the river width and flow do not vary, and the sediment effects are not considered, it is expected that in steeper sections, the water tends to flow faster and spend less time interacting with the riverbed. Therefore, its force has a smaller impact on the river bathymetry."*

- Lines 264-266 (was line 240): *"Physically, it can be understood that, in the river sections where the river width and slope do not vary, and the sediment influences are not considered, the increased flow has greater water force, which is correlated with a higher impact on the river bathymetry than smaller flow."*

- Apart from that, in Section 4 (Discussion) in the manuscript, we suggested further research to include formulas that consider sediment conditions when estimating the river bathymetry at lines 418-421: *"... Furthermore, the analysis framework in this research can be applied to a wide range of formulas, such as those that also consider the sediment impacts, that are used to estimate river bathymetries to represent rivers in the flood modelling. Future research about this can help to answer which formula contributes the most to the uncertainty in flood model outputs."*

**3.10 Question 10**

*Question:* L264-266: This is common knowledge and redundant.

*Answer:* We have removed this common knowledge at lines 289-291: *" ... a steeper or wider river typically becomes shallower, while an increase in the flow corresponds to a deeper river. Moreover, ..."*, and rewrote into: *"Overall, the variation in the river width corresponds to the largest variability in the river bathymetry followed by variations in the river flow and slope. Besides, ..."*

**3.11 Question 11**

*Question:* L280: Why are there smaller depths in smaller slopes (Figure 5c)? Is it near the coast?

205 *Answer:* To explain, in this area near the coast, the river depths are more correlated to the rise of the river width than the decrease of the river slope. We have explained and rewritten about this at lines 334-342 as below:

*"Furthermore, in the downstream reach, given the flat terrain, the increase in width outweighs the decrease in slope. Mathematically, the slope and width are in the denominator of both formulas, indicating their inverse relationships with the river bathymetries. Moreover, the slope drop (within 80 %) and its exponents (0.3 and 0.175 for the UF and CMR formulas)*

210 *are much smaller than the width increase (within 400 %) and its exponents (0.6 and 0.573 for the UF and CMR formulas). Consequently, the river bathymetries are affected by the increase in the river width than the decrease in the slope. Besides, as mentioned in Section 3.1, when the width starts increasing and the slope keeps decreasing, the river bathymetries of both formulas first converge, then diverge, with the UF bathymetries eventually exceeding the CMR bathymetries."*

**3.12 Question 12**

215 *Question:* Figure 5: It is better to indicate the panel number (a, b, c, . . . ) closer to the plot or inside the axes. Overall, the description of Figure 5 in the text is vague and hard to follow (L276-306).

*Answer:* We have moved the panel numbers (a), (b), (c) closer to the plots to be consistent with other panel numbers. We did not put them inside the axes as they will overlap with some information. We have also rewritten and added more information in the description of Figure 5 at lines 301-342 (was lines 276-306) for clarity and easily following as below:

[revised manuscript text omitted]

**3.13    Question 13**

*Question:* L307-322 and Figure 6c,d: What is the reason for the abrupt change in Cov of flow in the downstream of the river in Figure 6c,d?

*Answer:* The flow values were provided for each segment along the river. In this project, between the downstream (river

260     mouth) and upstream (where the Waikanae Water Treatment Plant gauge is), there are three flow values representing for three segments along the selected section of the river. They are 145.196, 145.978, and 146.194 cumecs, and the CoV of the bathymetry of these segments are 3.969, 4.315, and 4.351% for UF formula, and 3.792, 4.123, and 4.156% for CMR formula. The flow difference between the first and second segments is larger than the one between the second and third segments, and thus the differences in CoV values of the bathymetric data between these segments also follow a similar trend. This leads to

265     the abrupt change in color as we can see in Fig. 6c-d.

**3.14    Question 14**

*Question:* Figure 6: The authors should show meaningful values in the color bars. It is worthwhile to include the 1000 m mark in this figure.

*Answer:* We have added the 1000 m mark in the Figure 6. Also, for each dataset, the coefficient of variation (CoV) values

270     in the color bars show the variations in the simulated bathymetries. Apart from that, each row in the Figure 6 represents for the parameter that has its errors added, and each column represents for the formula used to calculate the bathymetry. This

informtation has been described in the caption: *"Variations in the simulated Waikanae River bathymetries due to associated error distributions in parameters: the Conceptual Multivariate Regression formula - (a) slope, (c) bank-full flow, (e) width, and (g) combined; the Uniform Flow formula - (b) slope, (d) bank-full flow, (f) width, and (h) combined.".*

Based on this figure, by comparing the ranges of CoV values between each row we can see that the variation in the simulated bathymetries increases between the slope, flow, and width datasets. This has been mentioned at line 345: *"In both formulas, the ranges of coefficients of variations increases between the slope, flow, and width datasets.".* Between two formulas, we can see that the first column (Conceptual Multivariate Regression formula) has darker colors than the second column (Uniform Flow formula). In other words, the variations in simulated bathymetries using the Conceptual Multivariate Regression formula is smaller than using the Uniform Flow formula. This has also been mentioned at lines 347-348: *"Moreover, the colors of the UF-formula river bathymetries are darker than those of the CMR-formula ones. This demonstrates the UF-formula bathymetries exhibit larger variability than those from the CMR formula.".*

To make it clearer, the labels of the colorbars are changed as below:

- We changed *"Coefficient of variation of SLOPE (%)"* into *"Coefficient of variation of simulated bathymetries of SLOPE dataset (%)"*

- We changed *"Coefficient of variation of FLOW (%)"* into *"Coefficient of variation of simulated bathymetries of FLOW dataset (%)"*

- We changed *"Coefficient of variation of WIDTH (%)"* into *"Coefficient of variation of simulated bathymetries of WIDTH dataset (%)"*

- We changed *"Coefficient of variation of COMBINED parameters (%)"* into *"Coefficient of variation of simulated bathymetries of COMBINED dataset (%)"*

**3.15  Question 15**

*Question:* Why did authors not use observed water levels to better understand the changes in river bathymetry? Using observed or satellite-based water surface elevations would help authors to understand the effect of the estimates.

*Answer:* We agree with the reviewer that the observed or satellite-based water surface elevations would help to further understand the effects of the estimated river bathymetry. However, we do not have this data, and thus we used the other available observed data as mentioned in the paper - the observed flood levels under point format. As mentioned in Section 3.3 (question 3) in this response, we have added this information at lines 245-251: *"In our research, we went further than Nguyen et al. (2025) by validating each flood simulation - MWSE with the observed data. Due to the lack of a thorough map of measured flood levels or satellite-based water surface elevations, we used the observed flood levels under point format provided by Wallace (2010). The Root Mean Square Error (RMSE) metric was harnessed for these validations. Locations of the observed data where the flood model predicted to be dry across all the simulations were removed to ensure the RMSE focuses only on*

305  ## 3.16  Question 16

*Question:* L360: The authors should clarify how the RMSEs were calculated.

    *Answer:* We haved described how the RMSEs were calculated in Section 2.5.2. at lines 245-251 (also as mentioned in Section 3.3 and 3.15 (questions 3 and 15) in this response): *"In our research, we went further than Nguyen et al. (2025) by validating each flood simulation - MWSE with the observed data. Due to the lack of a thorough map of measured flood levels*
310  *or satellite-based water surface elevations, we used the observed flood levels under point format provided by Wallace (2010). The Root Mean Square Error (RMSE) metric was harnessed for these validations. Locations of the observed data where the flood model predicted to be dry across all the simulations were removed to ensure the RMSE focuses only on predicted flooded regions and to avoid skewing the RMSE. We then visualised the distribution of RMSEs across simulations through side-by-side boxplot for comparison.".*

315      To make it clearer, we have added a note in the caption of Figure 11: *"RMSE distributions for predicted flood levels of eight datasets (slope-, flow-, width-, and combination-CMR and -UF datasets) compared to the January-2005 observed flood levels. The RMSEs were calculated using the method described in Section 2.5.2."*

**3.17  Question 17**

*Question:* L412-413: Then a question arises: why and where is the current method useful?
320      *Answer:* For situations where we lack the river bathymetric data and cannot collect or measure them for flood modelling, this study with its Monte Carlo assessment helps to show the sensitivity and understanding of the limitations and uncertainties involved. This has been rewritten and added at lines 410-421 as below:

    *"Our research went a step further than previous studies (Durand et al., 2008; Lee et al., 2018; Moramarco et al., 2019; Kechnit et al., 2024) to quantify the uncertainty in flood predictions due to the errors in the estimated river bathymetry. In*
325  *this research, we applied the Monte Carlo method to generate a large number of simulations to capture the typical variability in the flood predictions and included spatial variability in our method. Moreover, we not only considered associated error distributions in parameters collectively, but we also performed a sensitivity analysis to assess the impact of each parameter. Hence, for situations where we lack the river bathymetric data and cannot collect or measure them for flood modelling, the formulas in this study can be used with the Monte Carlo assessment here that shows the sensitivity and understanding of the*
330  *limitations and uncertainties involved. Furthermore, the analysis framework in this research can be applied to a wide range of formulas, such as those that also consider the sediment impacts, that are used to estimate river bathymetries to represent rivers in the flood modelling. Future research about this can help to answer which formula contributes the most to the uncertainty in flood model outputs."*

    Besides, in sensitivity analyses, the Monte Carlo framework allows the physical concepts to be used, so we can compare
335  the impacts of flood model inputs on the flood model outputs. In contrast, other methods like machine learning models can

only learn the relationships between the predictors and the outputs to make the predictions. They do not include the physical concepts to thoroughly explain the uncertainty propagation in flood models. Therefore, in such situations, the Monte Carlo framework should be applied.

For other cases, especially in flood risk management, where the uncertainty should be included but normally excluded, due to its computational expense (i.e. requirements of a large amount of simulations). Hence, a more computationally efficient method such as machine learning models should be considered. However, to develop such machine learning models, the Monte Carlo framework still plays an important role to produce the data for training and testing processes. Additionally, the framework can also serve as a reference to benchmark the machine learning models.

To enhance the clarity, lines 453-458 are rewritten: *"On the other hand, although applying the Monte Carlo framework to quantify this uncertainty is fully comprehensive, its requirement of a large amount of simulations can be seen as a computationally expensive problem. Due to this, the uncertainty quantification is not normally considered in the flood risk management. Hence, a more computational efficient method is essential. The machine learning approach, well-known for its more effective process to obtain the comparable results, is a good candidate which needs further investigation."*

**3.18    Question 18**

*Question:* Conclusion: The authors should improve the conclusion to present their findings and recommendations clearly and objectively, presenting them vaguely will substantially reduce the value of the manuscript.

*Answer:* We thank you very much for this suggestion to protect the value of our manuscript. We have rewritten the conclusion for clarity and easy comprehension at lines 460-492 as below:

*"Our research focused on quantifying the uncertainty in flood predictions due to the errors in parameters used to estimate the river bathymetries. We applied LISFLOOD-FP flood model within a Monte Carlo method to generate multiple flood simulations for the January-2005 Waikanae River flood event for analysis. We performed a sensitivity analysis on three estimated parameters (river slope, flow, and width) and two formulas (the UF and CMR formulas) to assess their error impacts on the flood predictions individually and collectively through the estimated river bathymetries.*

*We found that, among three parameters, the uncertainty in flood model outputs, when the errors were added into the river width, is higher than when the errors were added into the river flow, followed by the river slope. The combination of all of them was found to have the highest uncertainty. Between two formulas, the uncertainty in the flood predictions, especially in the flood depths and extents, when using the UF formula for estimating the river bathymetric data, is larger than using the CMR formula.*

*It is recommended that, instead of developing from scratch, the Monte Carlo framework used for the sensitivity analysis in this research should be applied to benchmark various formulas used to estimate the river bathymetries to represent rivers in flood modelling. Further study is necessary to confirm the broad applicability of the UF formula without river categorisation. Moreover, based on our results, the data collection process should focus on measuring the parameters (river width and flow) that have more significant impacts on the flood predictions if the resources are limited. Additionally, further investigations should also include the river Manning's n, and $\alpha$ and $\beta$ coefficients to perform a thorough sensitivity analysis.*

*Apart from that, we suggested another study to be implemented on many rivers with different features. In addition, further research should consider how different realistic sources of errors affect the flood predictions. Also, the impacts of grid resolution on the estimated river bathymetry and on the flood predictions should be focused in future study. Currently, to cover such uncertainty, a freeboard is often used, but it fails to cover the variation in the flood extent, and thus a further study is recommended to improve its effectiveness. Lastly, there is a need for simpler and faster methods than the Monte Carlo framework such as machine learning approaches to be included in flood risk management."*

**4   Extra changes**

We have edited some parts as below to make the content more accurate, consistent, and concise:

- We rewrote lines 130-134: *"In this study, LISFLOOD-FP (Bates et al., 2010; Neal et al., 2018), a 2D hydrodynamic model, was used to simulate the January-2005 flood event because it is well known for its computational efficiency and highly accurate flood model outputs (Nguyen et al., 2025). Also, it was calibrated for the Waikanae River in Nguyen et al. (2025). The DEM and Manning's n values, along with the flow information and tidal data mentioned above were used as input into this model."*

- We changed the minimum values along the river of parameter flow (Q) from 145.3 (cumec) to 145.2 (cumec) in Table 1 for correction.

- We rewrote lines 170-177 for more clarity: *"Before Monte Carlo simulation process, we explore the relationship between these parameters and the river bathymetries estimated by the UF and CMR formulas. At first, the mean value over the entire river section of each parameter is calculated as seen in Table 1. We then increase the mean value of each parameter, except for the river Manning's n, from 50% to 200% while keeping other parameters constant. This method allows us to observe how the river bathymetries from the two formulas are affected when a parameter is varied. The result analysis of this part is mentioned in Section 3.1.".*

- We changed the red color of *"errors"* into black color in Table 2.

- We rewrote lines 235-236: *"However, different to Nguyen et al. (2025), mMWDs and sdMWDs were not considered in the research due to no useful information."*